# Amphibian mast cells serve as barriers to chytrid fungus infections

Kelsey A Hauser[1], Christina N Garvey[1], Ryley S Crow[1],
Muhammad RH Hossainey[1], Dustin T Howard[1], Netra Ranganathan[1],
Lindsey K Gentry[2], Amulya Yaparla[1], Namarta Kalia[1], Mira Zelle[1],
Elizabeth J Jones[3], Anju N Duttargi[3], Louise A Rollins-Smith[4,5],
Carly R Muletz-Wolz[2], Leon Grayfer[1]*

[1]Department of Biological Sciences, The George Washington University, Washington, United States; [2]Center for Conservation Genomics, Smithsonian National Zoo & Conservation Biology Institute, Washington, United States; [3]Department of Oncology, Lombardi Comprehensive Cancer Center, Georgetown University, Washington, United States; [4]Departments of Pathology, Microbiology and Immunology, and of Pediatrics, Vanderbilt University School of Medicine, Nashville, United States; [5]Department of Biological Sciences, Vanderbilt University, Nashville, United States

*For correspondence:
leon_grayfer@gwu.edu

Competing interest: The authors declare that no competing interests exist.

**Abstract** Global amphibian declines are compounded by deadly disease outbreaks caused by the chytrid fungus, *Batrachochytrium dendrobatidis* (*Bd*). Much has been learned about the roles of amphibian skin-produced antimicrobial components and microbiomes in controlling *Bd*, yet almost nothing is known about the roles of skin-resident immune cells in anti-*Bd* defenses. Mammalian mast cells reside within and serve as key immune sentinels in barrier tissues like skin. Accordingly, we investigated the roles of *Xenopus laevis* frog mast cells during *Bd* infections. Our findings indicate that enrichment of *X. laevis* skin mast cells confers anti-*Bd* protection and ameliorates the inflammation-associated skin damage caused by *Bd* infection. This includes a significant reduction in infiltration of *Bd*-infected skin by neutrophils, promoting mucin content within cutaneous mucus glands, and preventing *Bd*-mediated changes to skin microbiomes. Mammalian mast cells are known for their production of the pleiotropic interleukin-4 (IL4) cytokine and our findings suggest that the *X. laevis* IL4 plays a key role in manifesting the effects seen following cutaneous mast cell enrichment. Together, this work underscores the importance of amphibian skin-resident immune cells in anti-*Bd* defenses and illuminates a novel avenue for investigating amphibian host–chytrid pathogen interactions.

## eLife assessment

This **important** study reveals the role of skin-resident mast cells in amphibians in mediating antimicrobial responses. The data are **compelling** and highlight species-specific biology that can cross-inform human mast cell biology in a species that does not rely on IgE as a primary mechanism for antimicrobial skin responses.

## Introduction

Catastrophic declines of hundreds of amphibian species across six continents have been causally linked to the chytrid fungi, *Batrachochytrium dendrobatidis* (*Bd*) and *Batrachochytrium salamandrivorans* (*Bsal*) (*Martel et al., 2014*; *Scheele et al., 2019*). Motile *Bd* zoospores readily colonize

keratinized skin of adult amphibians and keratinized mouthparts of tadpoles (*Van Rooij et al., 2015*). *Bd* skin infections culminate in chytridiomycosis, ultimately disrupting the function of this respiratory and barrier tissue (*Van Rooij et al., 2015*; *Van Rooij et al., 2010*; *Voyles et al., 2009*). Effectively combating chytrid infections requires a holistic understanding of amphibian cutaneous immune defenses. Research efforts up to this point have focused on antifungal capacities of amphibian skin-produced antimicrobial peptides (AMPs) (*Rollins-Smith, 2023*; *Varga et al., 2018*); commensal antifungal products (*Varga et al., 2018*; *Jani et al., 2021*; *Woodhams et al., 2023*); antifungal properties of mucus (*Woodhams et al., 2023*), alkaloids (*Rodríguez et al., 2017*), and lysozymes (*Grogan et al., 2018*; *Rollins-Smith et al., 2009*); as well as the roles of antibodies (*Van Rooij et al., 2010*; *Ramsey et al., 2010*). Many of these studies suggest that *Bd* exposure can elicit some immune protection through these mechanisms (*McMahon et al., 2014*). However, the contribution of skin-resident immune cells to amphibian anti-*Bd* defenses remains almost entirely unexplored.

Mammalian mast cells serve as sentinels of mucosal and connective tissues, including in barrier tissues like skin, where they maintain homeostasis and regulate immune responses (*Sobiepanek et al., 2022*). Other granulocyte-lineage cells such as neutrophils are generally not found in healthy tissues and only extravasate into sites of inflammation (*Hauser et al., 2023*). Consequently, mast cells are among the first immune cells to recognize and respond to skin-infiltrating pathogens. When activated, mast cells release pre-formed and *de novo*-generated immunomodulatory compounds that may serve to elicit, exacerbate, or ameliorate inflammatory responses (*Sobiepanek et al., 2022*). One of these mast cell-produced mediators, the interleukin-4 (IL4) cytokine dampens inflammation and promotes tissue repair (*Allen, 2023*). Cells bearing hallmark mast cell cytology have been reported across a range of non-mammalian species (*McNeil et al., 2007*; *Abraham and St John, 2010*), including amphibians (*Baccari et al., 2011*). Notably, the principal mast cell growth factor, stem cell factor (SCF, KIT ligand), is required for mast cell differentiation and survival (*Ito et al., 2012*) and is expressed by all vertebrates examined to date.

Here, we combine comprehensive *in vitro* and *in vivo* approaches to define the roles of amphibian (*Xenopus laevis*) mast cells during *Bd* infections. Our results provide compelling evidence that skin-resident immune cells contribute to anti-*Bd* defenses.

## Results

### Frog mast cells possess archetypal mast cell cytology and transcriptional profiles

We produced *X. laevis* recombinant (r)SCF and used this reagent to generate mast cell cultures from bone marrow-derived myeloid precursors (*Yaparla et al., 2016*). Mast cells were compared to bone marrow-derived neutrophilic granulocytes (hereafter referred to as 'neutrophils'), differentiated using a recombinant *X. laevis* colony-stimulating factor-3[22] (rCSF3, i.e., granulocyte colony-stimulating factor [GCSF]). While the neutrophil cultures were comprised of cells with hyper-segmented nuclei and neutral-staining cytoplasms (*Figure 1A*), the mast cell cultures consisted predominantly of mononuclear cells with basophilic cytoplasm (*Figure 1B*). We confirmed the granulocyte-lineage of *X. laevis* mast cells using specific esterase (SE) staining (*Figure 1D*). As expected, *X. laevis* neutrophils were also SE-positive (*Figure 1C*). Mast cell and neutrophil morphology was further explored with electron microscopy (*Figure 1E–H*). Scanning electron microscopy (SEM) imaging demonstrated that *X. laevis* mast cells possess extensive folding of their plasma membranes (*Figure 1F*). This mast cell-characteristic membrane ruffling appeared as projections resembling pseudopods via transmission electron microscopy (TEM), which further revealed electron-dense heterogeneous granules, few mitochondria, and round to elongated nuclei (*Figure 1H*) typical of mammalian mast cells (*Espinosa-Riquer et al., 2020*). *X. laevis* neutrophils also exhibited pronounced membrane ruffling (*Figure 1E*) but strikingly distinct intracellular appearance including multilobed nuclei (*Figure 1G*).

*X. laevis* mast cells and neutrophils exhibited distinct transcriptional profiles of immune-related genes including those encoding lineage-specific transcription factors, immune receptors, downstream signaling components and adhesion molecules, as well as non-immune genes (*Figure 1I*). Frog mast cells and neutrophils each expressed greater levels of lineage-specific transcription factors associated with mammalian mast cell (*gata1*, *gata2*, and *mitf*) (*Cildir et al., 2021*) and neutrophil (*cebp* family members) (*Paudel et al., 2022*) counterparts, respectively (*Figure 2A*). Notably, mast cells expressed

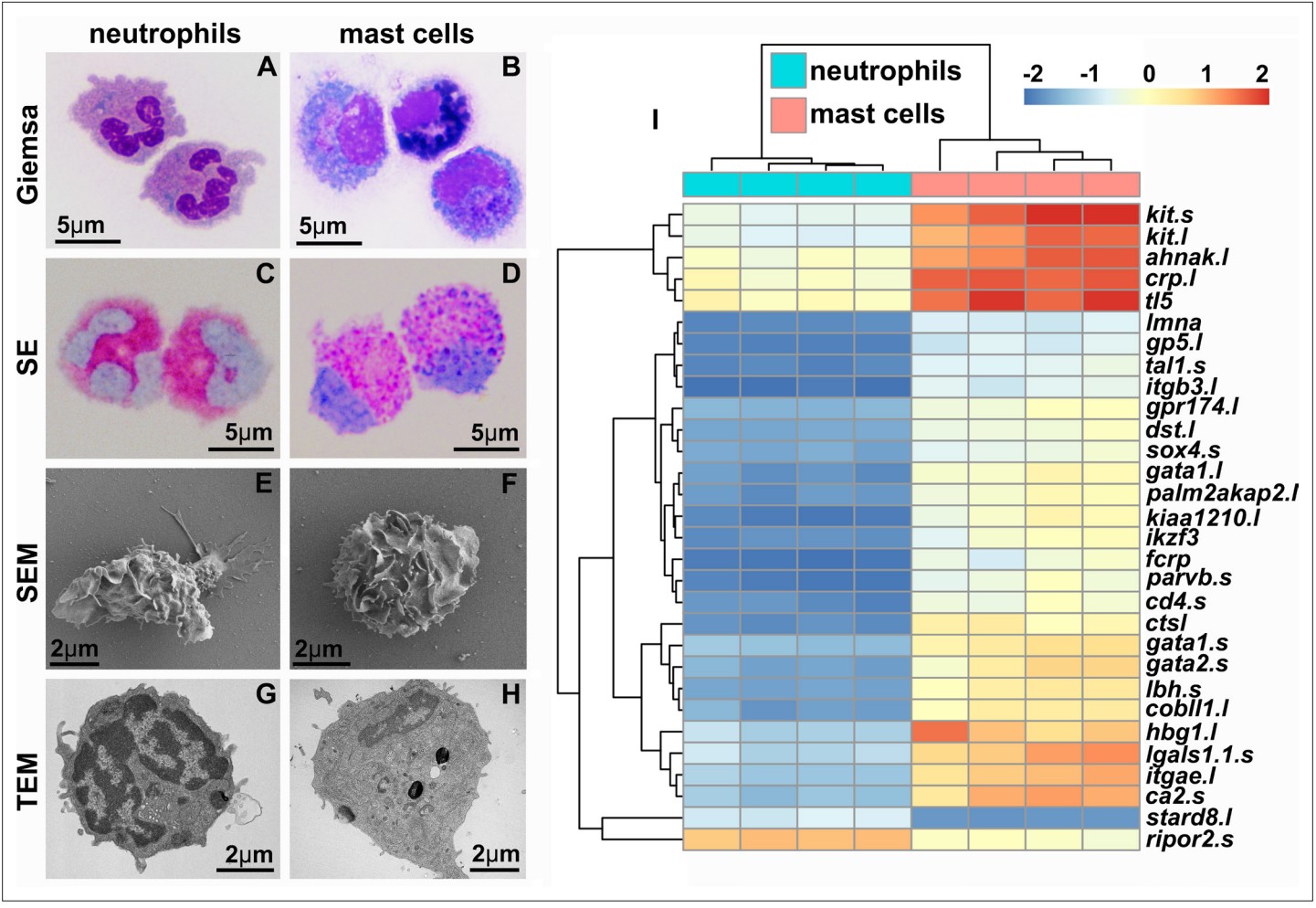

**Figure 1.** *X. laevis* bone marrow-derived mast cells possess archetypal mast cell cytology and transcriptional profiles. Neutrophils (**A, C, E, G**) and mast cells (**B, D, F, H**) were stained with Giemsa (**A, B**) and Leder to visualize specific esterase activity (SE) (**C, D**) or imaged with scanning and transmission electron microscopy (SEM: **E, F** and TEM: **G, H**). (**I**) Heat map of the top 30 differentially expressed genes (DEGs) identified with RNA sequencing analyses of *X. laevis* mast cell (N = 4) and neutrophil (N = 4) cultures. Log$_2$fold change in expression represented as color scale.

greater levels of enzyme and cytokine genes associated with tissue remodeling (carboxypeptidase-3 [*cpa3*]; *Atiakshin et al., 2022*), immune suppression (indoleamine 2,3 dioxygenase-1 [*ido1*]; *Baban et al., 2009*), and amelioration of cutaneous inflammation (leukemia inhibitory factor [*lif*]; *Banner et al., 1998*; *Figure 2B*). Conversely, neutrophils expressed predominantly proinflammatory enzymes and cytokine genes such as leukotriene 4 hydrolase (*lta4h*; *Figure 2B*) and tumor necrosis factor alpha (*tnfa*, *Figure 2B*). In addition, mast cells and neutrophils each had greater expression of genes encoding their respective growth factor receptors, *kit* and *csf3r* (*Figure 2B*).

## Enriching frog cutaneous mast cells confers protection against *Bd*

Although all granulocyte-lineage cells possess specific esterase activity, mast cells are the predominant mononuclear granulocytes to reside in vertebrate tissues (*Agis et al., 1996*). Therefore, we selectively enriched mast cells in *X. laevis* skin via subcutaneous rSCF administration (note SE-stained cells indicated by arrows in r-ctrl-injected skins, *Figure 3A*, versus r-SCF-injected skins, *Figure 3B*). We confirmed SE-positive cells in rSCF-treated skins also possessed round-oval nuclei (*Figure 3C*). Maximum mast cell enrichment was observed 12 hr post injection of rSCF (*Figure 3D*).

We next examined the consequences of enriching frog skin mast cells on *Bd* infection outcomes. To this end, *X. laevis* were subcutaneously administered with rSCF or a recombinant control (r-ctrl), infected with *Bd*, and the skin fungal loads assessed at 10 and 21 days post infection (dpi). At 10 dpi, skin mast cell-enriched *X. laevis* possessed significantly lower *Bd* loads than r-ctrl-administered

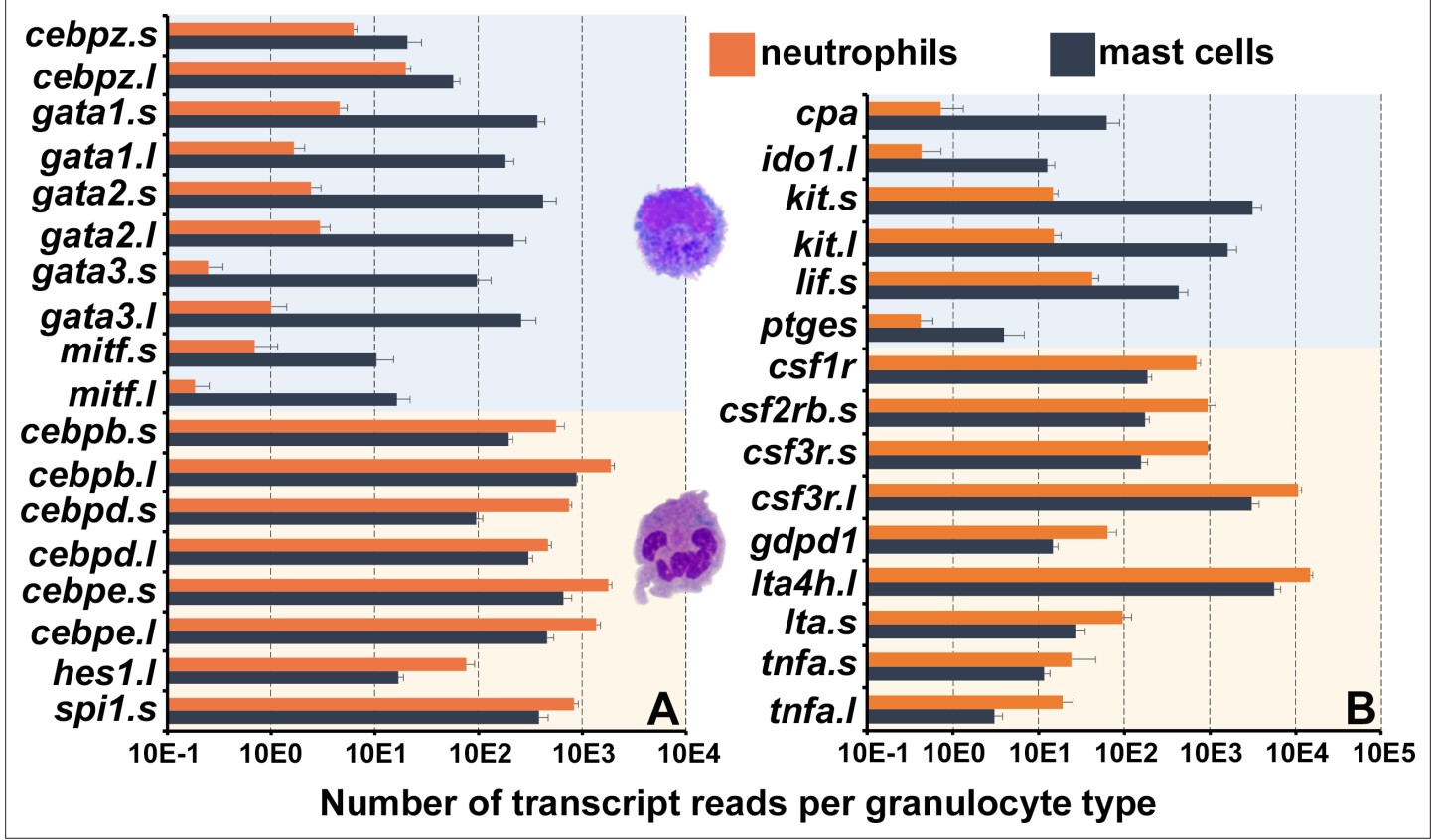

**Figure 2.** Frog mast cells and neutrophils possess gene profiles similar to their mammalian counterparts. The differentially expressed genes from the RNA sequencing analyses of *X. laevis* mast cells and neutrophil cultures were profiled for those encoding (**A**) transcription factors associated with mast cell- or neutrophil-specific lineages and (**B**) granulocyte antimicrobial components and growth factor receptor genes. All depicted genes were significantly differentially expressed between the two populations, N = 4 per group.

animals (*Figure 3E*). By 21 dpi, both r-ctrl and rSCF-administered groups possessed substantially greater *Bd* loads, although the mast cell-enriched animals continued to show significantly lower skin fungal loads (*Figure 3E*).

Mammalian mast cells may be labeled *in situ* with avidin to visualize the heparin-containing granules (*Tharp et al., 1985*). Using this approach, we confirmed that frog bone marrow-derived mast cells likewise possess heparin-containing granules (*Figure 4A and B*). We next used this staining approach to visualize and enumerate mast cells in the skins of control, mast cell-enriched mock- and *Bd*-challenged animals (*Figure 4C–I*). We observed cutaneous frog mast cell enrichment progressively waned with time in mock-infected animals, and by 21 days post mock infection, these animals had skin mast cell numbers comparable to r-ctrl-injected, mock-infected animals (*Figure 4C and D*). Conversely, compared to mock-infected control animals, the rSCF-injected, *Bd*-challenged frogs maintained significantly more cutaneous mast cells numbers throughout the 21-day infection study (*Figure 4C and E–H*). Interestingly, r-ctrl-injected, *Bd*-infected animals exhibited significantly increased skin mast cell numbers at 10 and 21 days of infection compared to uninfected control frogs (*Figure 4C and I*). Most of these mast cells were observed in the epidermal layer, spreading out between the skin epithelial cells (*Figure 4G*).

## Mast cells protect frogs from *Bd*-elicited inflammation

To explore potential mechanisms of mast cell-mediated protection against *Bd,* we compared the gene expression profiles of r-ctrl- and rSCF-administered, *Bd*-infected frog skins at 21 dpi. Among the top differentially expressed genes, we noted mast cell-enriched, *Bd*-infected skins possessed more transcripts for genes associated with cutaneous strength and integrity (*lamc2*) (*Kiritsi et al., 2013*), suppression of cell migration (*b3gnt3.1*) (*Ho et al., 2013*), as well as ion and nutrient flow (*gjb3l*)

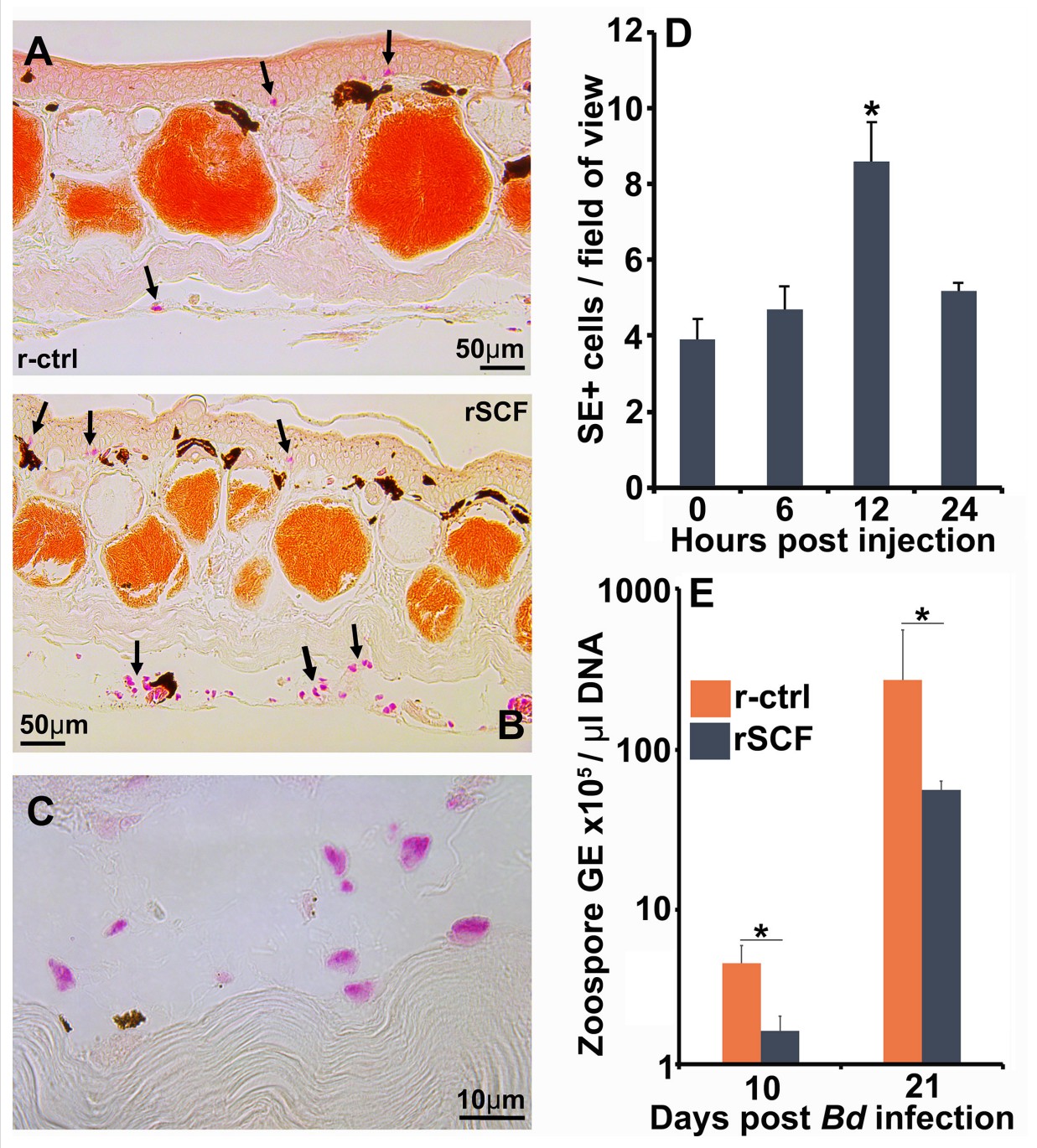

**Figure 3.** Enriching frog cutaneous mast cells lowers *Bd* loads. Representative images of specific esterase (SE) stained (**A**) control and (**B**) mast cell-enriched skin 12 hr post injection (hpi). (**C**) We confirmed the enriched population was composed of mono-morphonuclear cells. (**D**) Mast cell enrichment was optimized across several time points by quantifying SE-positive cells per field of view under ×40 magnification. Results represent means ±SEM from three animals per time point (two experimental repeats). (**E**) Mast cell-enriched and control dorsal skins were collected from *X. laevis* 10 and 21 dpi. *Bd* loads are represented as the number of zoospore genomic equivalents (GE) × $10^5$ per μL of total input DNA. Time points were analyzed independently. Results represent means ±SEM from seven animals per experimental group (N = 7). Asterisks indicate significance: p<0.05 by (**D**) one-way ANOVA with Tukey post hoc analysis or (**E**) Student's *t*-test.

(*Chang et al., 2003*; *Figure 5A*). Moreover, mast cell-enriched *Bd*-challenged skin exhibited greater expression of genes associated with protection of the mucosa and epithelial healing (*ttf3.6s*) (*Taupin and Podolsky, 2003*) and mucus production (*duoxa1.s* [*Fischer, 2009*], *gabrp* [*Wang et al., 2021*]; *Figure 5A*). In striking contrast, skins from control *Bd*-infected frogs revealed greater expression of

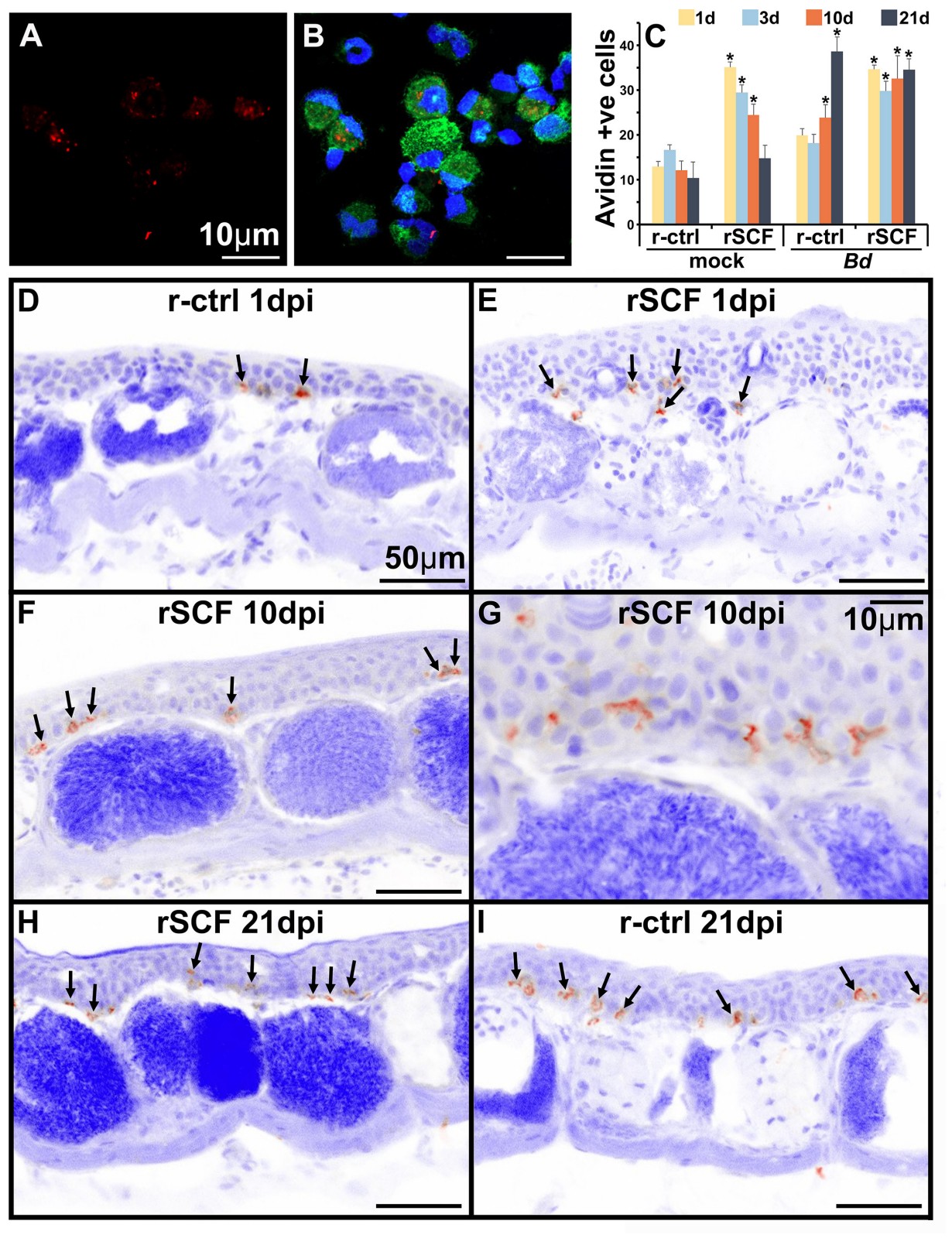

**Figure 4.** Heparin content and skin localization of frog mast cells. Frogs were administered r-ctrl or rSCF subcutaneously, mock or *Bd*-challenged, and their skins examined after 1, 3, 10, and 21 days post infection (dpi). (**A, B**) Representative images (cultures derived from five individual frogs) of bone marrow-derived frog mast cells, stained with fluorescently labeled avidin to visualize heparin-containing granules (avidin: red; nuclei: blue; actin: green). (**C–I**) Skin tissue from control (r-ctrl) or mast cell-enriched (rSCF), mock- (not shown) and *Bd*-infected (**D–I**) *X. laevis* were stained with

*Figure 4 continued on next page*

*Figure 4 continued*

fluorescently labeled avidin to visualize mast cells therein (N = 6 animals per treatment group). Images were inverted in ImageJ for greater contrast and visibility. (**C**) Heparin-containing mast cells were enumerated and depicted as means ± SEM of heparin-positive cells per field of view, N = 6 animals per treatment group. Asterisks indicate statistical significance from r-ctrl: p<0.05. Representative images of heparin-containing mast cells in the skins of (**D**) r-ctrl animals 1 dpi; (**E**) rSCF-administered frogs 1 dpi; (**F, G**) rSCF-administered frogs 10 dpi; (**H**) rSCF-administered frogs 21 dpi; and (**I**) r-ctrl-treated animals 21 dpi with *Bd*.

genes associated with leukocyte infiltration and inflammation (e.g., *ccl19*, *cxcl16*, *adamts13*, *csf3r*; *Figure 5A*). These transcriptional profiles were supported by our histological observations wherein control (r-ctrl), *Bd*-infected skins exhibited hyperkeratosis, epidermal hyperplasia, jagged stratum corneum, and extensive leukocyte infiltration (*Figure 5B*), while mast cell-enriched *Bd*-infected tissues appeared considerably less afflicted by these pathologies (*Figure 5C*). Quantification of *Bd*-infected skin thickness confirmed that mast cell-enriched animals possessed significantly less thickened epidermal skin compared to control (r-ctrl), *Bd*-infected animals (*Figure 5D*).

Because mast cell-enriched frog skins had greater expression of genes associated with mucosal tissue integrity and mucus production (*Figure 5A*), we also investigated whether the anti-*Bd* protection identified in mast cell-enriched skins could be due at least in part to differences in mucus production. Interestingly, cutaneous mucus glands of mast cell-enriched, mock- and *Bd*-infected frogs were significantly more filled than those of mock- and *Bd*-infected control animals (*Figure 5B, C, and E*).

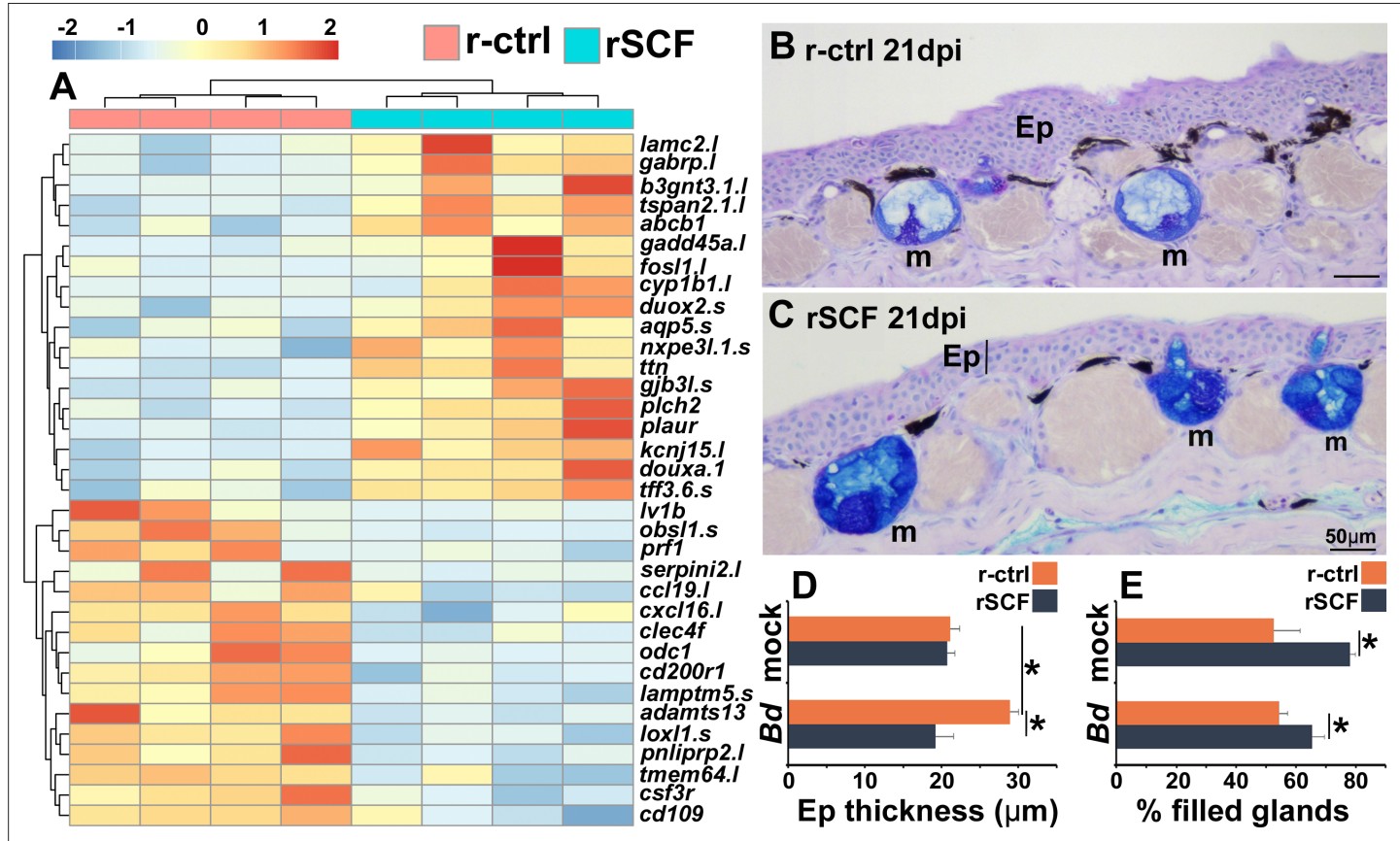

**Figure 5.** Consequences of cutaneous mast cell enrichment. (**A**) RNAseq analysis of skin tissue from control (r-ctrl) or mast cell-enriched (rSCF) *Bd*-infected *X. laevis* at 21 days post infection (dpi). Heat map of the top 30 differentially expressed genes (DEGs), numbers matched to colors represent log$_2$ fold change in expression, N = 4 r-ctrl-treated, *Bd*-infected and 4 rSCF-treated, *Bd*-infected skin samples. (**B, C**) Representative images of control and mast cell-enriched, *Bd*-infected skins, 21 dpi, demonstrating differences in epidermal thickening and mucus gland filling. Mucin content was visualized in cutaneous mucus glands with Alcian Blue/PAS stain. Mucus glands are denoted by 'm', and epithelia are denoted by 'Ep'. ImageJ software was used to determine the means ± SEM of (**D**) skin epithelial thickness and (**E**) percent mucus gland filling (N = 6). Asterisks indicate significance: p<0.05 by one-way ANOVA with Tukey post hoc analysis.

## Cutaneous neutrophil enrichment results in increased *Bd* fungal loads

Neutrophils are one of the first leukocytes to infiltrate infected tissues, typically amplifying inflammation (*Wang, 2018*). All vertebrate neutrophils depend on CSF3 for their differentiation and function (*Roberts, 2005*), and our previous work has confirmed that this is true of frog neutrophils, which intuitively also exhibit high expression the CSF3 receptor (*csf3r*) (*Koubourli et al., 2017*; *Yaparla et al., 2016*). Because *csf3r* expression was markedly elevated in control over mast cell-enriched skins of infected frogs (*Figure 5A*), we examined the neutrophil content in the skins of these animals over the 21-day course of *Bd* infection (*Figure 6A–E*). To this end, we used *in situ* hybridization analysis of skin-myeloperoxidase (*mpo*), a marker of neutrophils (*Buchan et al., 2019*). While the skins of r-ctrl- and rSCF-administered, mock-infected frogs contained relatively few neutrophils (*Figure 6A and E*), the skins of r-ctrl-injected, *Bd*-infected animals had significantly elevated neutrophil numbers (*Figure 6B, C, and E*). Conversely, at both examined time points (10, 21 dpi), the skins of mast cell-enriched frogs possessed neutrophil levels similar to those seen in the uninfected animals (*Figure 6D and E*).

We next assessed the consequence of enriching frog skins for neutrophils via subcutaneous rCSF3 administration. We confirmed neutrophil enrichment peaked 12 hp rCSF3 injection (*Figure 6—figure supplement 1*) and resulted in a thickened epidermis in comparison with r-ctrl-injected skins of otherwise healthy animals (i.e., no *Bd*; *Figure 6F and G*). When challenged with *Bd*, frogs with neutrophil-enriched skin possessed significantly greater *Bd* loads than control frogs (*Figure 6H*), suggesting that neutrophil-mediated inflammation may be exacerbating *Bd* infections.

## Enrichment of frog skin mast cells alters skin microbial composition

We observed substantial differences in skin microbiomes, including changes in bacterial composition and richness as well as relative abundances of *Bd*-inhibitory bacteria (*Figure 7*, *Figure 7—figure supplement 1*). A total of 1645 bacterial amplicon sequence variants (ASVs) were identified from 20 bacterial phyla, 7 of which were predominant (*Figure 7A*). Of these, *Verrucomicrobiota* were only present on uninfected animals, whereas *Acidobacteriota* was only seen after 21 dpi on both control and mast cell-enriched, infected animals (*Figure 7A*).

At 10 dpi, mast cell enrichment resulted in a nominal shift in community composition compared to control frogs (*Figure 7B*). Notably, while control, *Bd*-infected animals exhibited a drastic shift in community composition, mast cell-enriched animals possessed substantially less deviated community composition (*Figure 7B*), suggesting that these cells are somehow counteracting the adverse effects of *Bd* on the *X. laevis* cutaneous microbiome communities. These mast cell-mediated effects persisted to 21 dpi (*Figure 7—figure supplement 1A*).

At 10 dpi, mast cell-enriched and mock-infected frogs possessed significantly greater abundance of *Bd*-inhibitory bacteria such as *Chryseobacterium* sp. (*Woodhams, 2015*), compared to control, mock-infected animals (*Figure 7C*). This suggests mast cells may promote skin flora composition that is more antifungal. Control (non-enriched) *Bd*-infected frogs possessed significantly greater abundance of *Bd*-inhibitory bacteria than all other treatment groups (*Figure 7C*). While mast cell-enriched, *Bd*-infected frogs had lower abundance of *Bd*-inhibitory bacteria than control infected frogs, they possessed higher abundance of inhibitory taxa than uninfected control animals (*Figure 7C*). The *Bd*-inhibitory bacteria seen in greater abundance on mast cell-enriched, *Bd*-infected animals included *Roseateles* sp., *Flavobacterium* sp., and *Kaistia* sp. We did not see significant differences in *Bd*-inhibitory bacteria across treatment groups at 21 dpi (*Figure 7—figure supplement 1B*).

Mast cell-enriched uninfected frogs exhibited increased skin bacterial richness at 10 dpi (*Figure 7D*). While control *Bd*-infected animals exhibited significantly reduced skin microbial richness, mast cell-enriched *Bd*-infected frogs did not exhibit such a reduction in bacterial richness (*Figure 7D*), supporting the idea that mast cells may be counteracting the adverse effects of *Bd* on skin microbiome composition.

In addition to serving as a selective substratum for commensal microbes, amphibian skin-produced mucus may also offer antifungal protection (*Woodhams et al., 2023*). However, we found no significant differences in direct *Bd*-killing capacities of mucus isolated from mock- or *Bd*-challenged control or mast cell-enriched frogs (*Figure 7—figure supplement 2*).

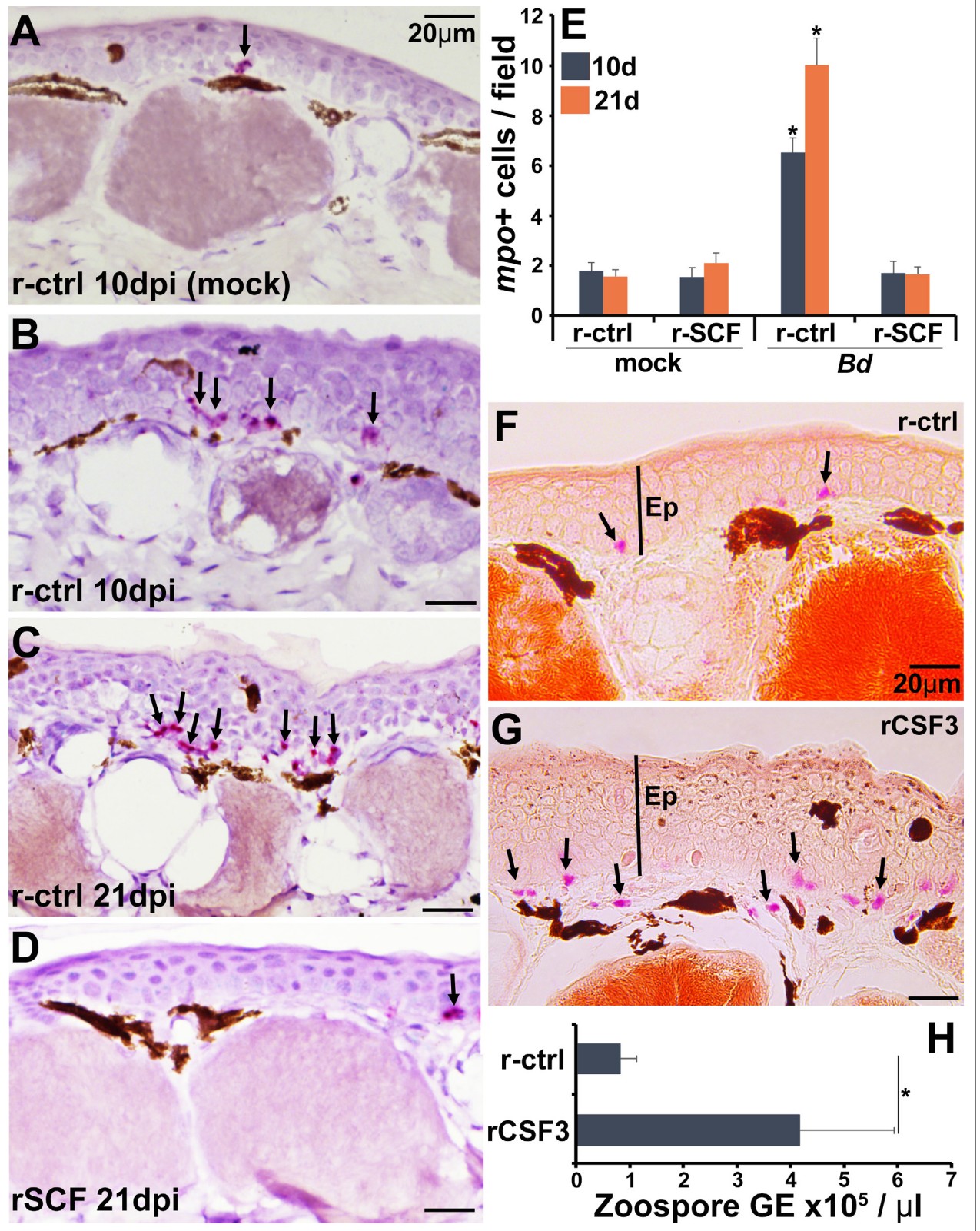

**Figure 6.** Consequences of cutaneous neutrophil enrichment. (**A–E**) Frogs were administered with r-ctrl or rSCF, mock- or *Bd*-infected, and examined *in situ* for neutrophil content via by RNAScope analyses of myeloperoxidase (*mpo*) transcripts. Representative images of (**A**) r-ctrl-injected frog skins 10 days post mock infection; (**B**) r-ctrl-injected frog skins 10 days post *Bd* infection; (**C**) r-ctrl-injected frog skins 21 days post *Bd* infection; and (**D**) r-SCF-administered frog skins 21 days post *Bd* infection. (**E**) Means ± SEM of *mpo*-positive neutrophils per field of view of r-ctrl- and rSCF-administered,

*Figure 6 continued on next page*

*Figure 6 continued*

mock- or *Bd*-challenged frog skins, 10 or 21 days post infection (dpi), (N = 6). Asterisks indicate significance from control: p<0.05 by one-way ANOVA with Tukey post hoc analysis. Representative specific-esterase staining of skin tissues from frogs were administered with (**F**) r-ctrl or (**G**) rCSF3, N = 4. Arrows denote specific esterase-positive cells. Ep: epithelium. (**H**) *Bd* loads in control and neutrophil-enriched skin tissues 7 dpi, N = 6. Asterisks indicate significance: p<0.05 by one-way ANOVA with Tukey post hoc analysis.

The online version of this article includes the following figure supplement(s) for figure 6:

**Figure supplement 1.** Enrichment of neutrophils in frog skins.

## Frog mast cells alter skin antimicrobial peptide gene expression

Amphibians rely heavily on skin-produced AMPs for antifungal protection (*Patocka et al., 2019*), and mast cells produce antimicrobial AMPs (*Di Nardo et al., 2008*). We thus examined whether mast cells could be sources of such AMPs during *Bd* infections. As anticipated, *Bd*-challenged mast cells, but not *Bd*-challenged neutrophils, upregulated their expression of the AMP-encoding genes, PGLa (*pgla*) and magainin (*mag*, *Figure 8A*).

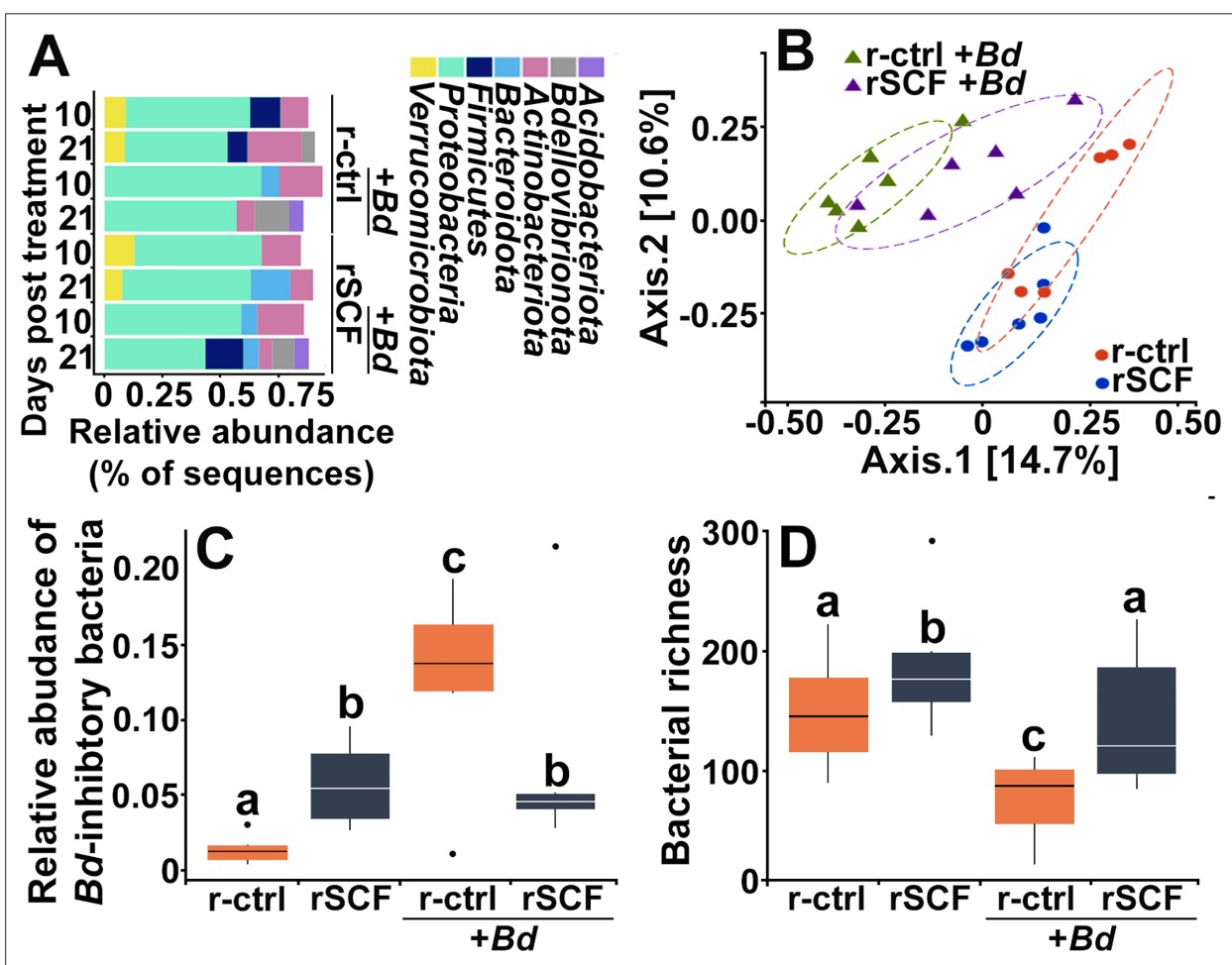

**Figure 7.** Cutaneous mast cells protect skin microbial communities. Control (r-ctrl-injected) or mast cell-enriched (rSCF-injected) *X. laevis* were mock-infected or challenged with *Bd* for 21 days. (**A**) Microbial phyla distribution across groups. Low-abundance phyla (<5% relative abundance are not shown). At 10 days post infection (dpi) (**B**), community composition (Jaccard distances shown with 80% confidence ellipses) differed among all treatments. (**C**) Relative abundance of *Bd*-inhibitory bacteria and (**D**) bacterial richness were examined in control and mast cell-enriched frogs, 10 days post *Bd* or mock challenge. Letters above bars indicate statistically different groups.

The online version of this article includes the following figure supplement(s) for figure 7:

**Figure supplement 1.** Mast cell enrichment protects frogs from *Bd*-mediated changes to skin microbiomes.

**Figure supplement 2.** Mucosomes from mast cell-enriched frogs do not confer *Bd* killing.

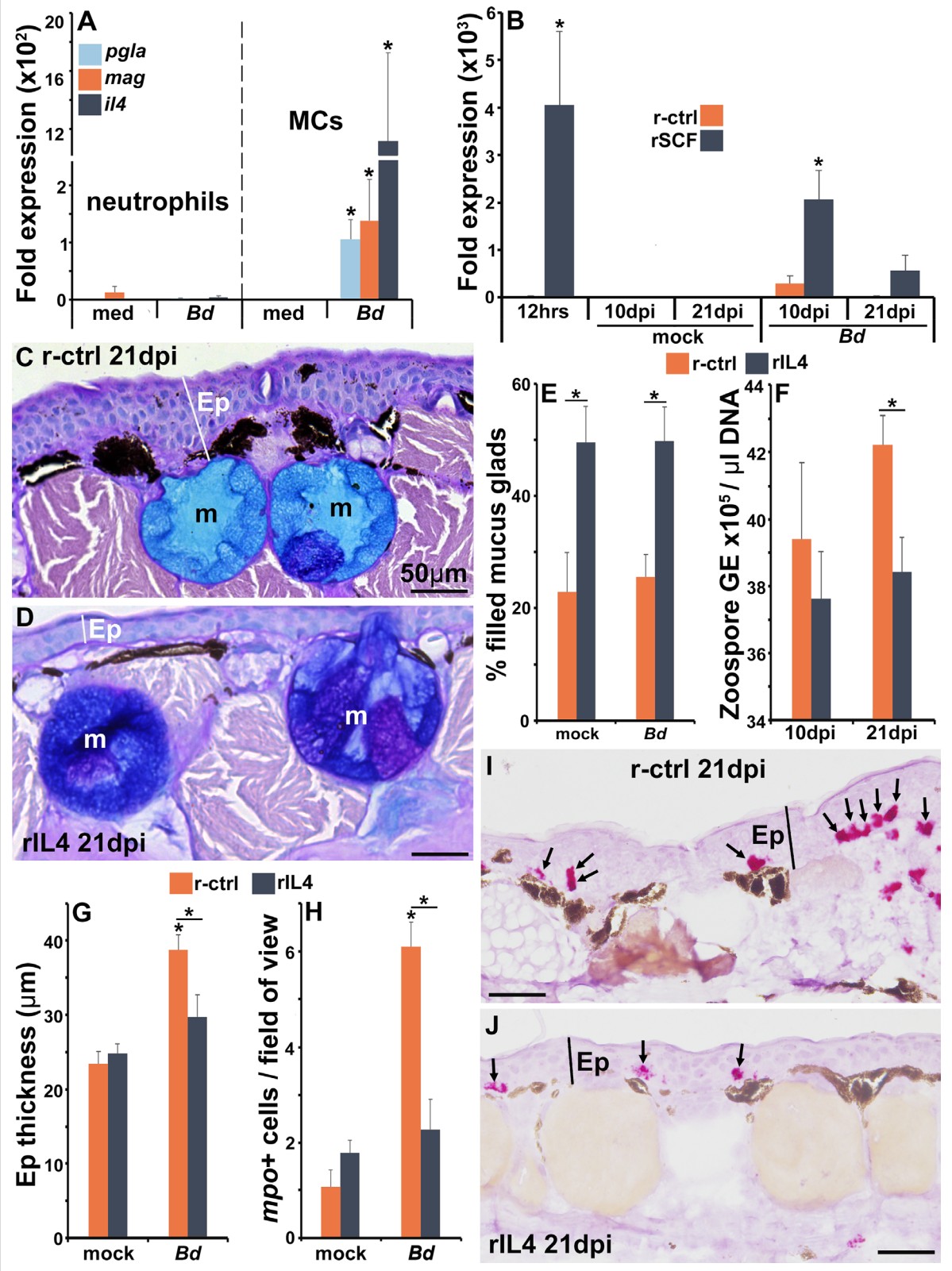

**Figure 8.** The roles of IL4 in mast cell-mediated skin anti-*Bd* protection. (**A**) Mast cells (MCs) and neutrophils derived from bone marrow of six individual frogs (N = 6) were co-cultured with *Bd* (five fungal cells per granulocyte) for 6 hr prior to gene expression analyses of the antimicrobial peptide genes PGLa (*pgla*) and magainin (*mag*) or interleukin-4 (*il4*). (**B**) *Il4* gene expression in skins of control and mast cell-enriched, mock- and *Bd*-infected animals, N = 6. Representative images of frogs administered with (**C**) r-ctrl or (**D**) rIL4 and infected with *Bd* for 21 days, N = 7. Ep: epidermis; m: mucus gland.

*Figure 8 continued on next page*

*Figure 8 continued*

Means ± SEM of (**E**) percent mucus gland filling, (**F**) skin *Bd* loads, (**G**) epidermal thickness, and (**H**) *mpo*-positive neutrophils, per field of view of r-ctrl- or rIL4-administered, mock- or *Bd*-challenged frog skins 21 days post infection (dpi) (N = 7). Representative images of *mpo*-positive neutrophils in (**I**) control and (**J**) rIL4-treated frog skins, 21 dpi. Ep: epidermis; arrows: mpo-positive neutrophils. Asterisks indicate significance: p<0.05 by one-way ANOVA with Tukey post hoc analysis.

The online version of this article includes the following figure supplement(s) for figure 8:

**Figure supplement 1.** Expression of antimicrobial peptides in mast cell-enriched, *Bd*-infected frog skins.

**Figure supplement 2.** IL4 elicits hallmark gene expression but does not protect *Bd*-infected frogs.

To follow up these observations *in vivo*, we examined *pgla* and *mag* gene expression in the skins of mast cell-enriched and *Bd*-challenged animals. Compared to control (r-ctrl) animals, mast cell-enriched frogs did not have elevated mRNA levels of *mag* or *pgla* after 12 hr of rSCF administration (*Figure 8—figure supplement 1A and B*). Conversely, after 10 days of mock infection, mast cell-enriched animals possessed significantly greater skin expression of *mag* and *pgla* than control animals (*Figure 8—figure supplement 1A and B*). We did not see significant differences in skin *mag* or *pgla* gene expression between control and mast cell-enriched, *Bd*-challenged frogs after 10 dpi (*Figure 8—figure supplement 1*). By 21 days of challenge, the rSCF-administered mock-infected frogs possessed lower gene expression levels of both AMPs (significantly so for *pgla*, *Figure 8—figure supplement 1A and B*), possibly due to some sort of compensatory effect. At 21 dpi, mast cell-enriched frog skins had greater *pgla* expression than control frogs, though these differences did not reach statistical significance (*Figure 8—figure supplement 1B*).

## Frog mast cell-expressed interleukin-4 confers anti-*Bd* protection

Mammalian mast cells are recognized as potent producers of the pleotropic anti-inflammatory cytokine, interleukin-4 (*il4*) (*Babina et al., 2016*). Notably, *X. laevis* mast cells challenged *in vitro* with *Bd* significantly upregulated their *il4* gene expression, whereas almost no *il4* expression was detected from either unstimulated or *Bd*-challenged neutrophils (*Figure 8A*).

When we examined the expression of *il4* in the skin of control and mast cell-enriched animals, we found that 12 hr following rSCF administration (time of *Bd* challenge), the mast cell-enriched frog skins possessed significantly greater transcripts of *il4* than control animals (*Figure 8B*). By 10 and 21 dpi, both control and mast cell-enriched, mock-infected animals possessed baseline skin *il4* expression (*Figure 8B*). At 10 days of *Bd* infection, control animals possessed increased (not significantly) skin *il4* gene expression whereas mast cell-enriched frog skins had significantly elevated *il4* transcript levels (*Figure 8B*). By 21 dpi, control, *Bd*-infected animals had baseline skin *il4* expression while the mast cell-enriched animals had elevated, albeit not significant expression of this cytokine gene in their skins (*Figure 8B*).

To examine whether IL4 could confer the anti-*Bd* effects seen following frog skin mast cell enrichment, we produced this *X. laevis* cytokine in recombinant form (rIL4) and confirmed that subcutaneous injection of rIL4 augmented expression of genes known to be activated by the mammalian IL4 (*Bhattacharjee et al., 2013*) (*cd36*, metalloproteinase inhibitor 3-*timp3*, and monoamine oxidase A-*maoa*; *Figure 8—figure supplement 2A*).

We next administered frogs subcutaneously with rIL4 or r-ctrl, challenged them with *Bd* and examined the key parameters affected by skin mast cell enrichment, including skin mucus gland filling, skin *Bd* loads, and reduction in *Bd*-associated skin inflammation. When we examined the mucus content of control and rIL4-administered animals, we found that frogs treated with rIL4 possessed greater skin mucus gland filling than control animals, irrespective of mock- or *Bd*-challenge (*Figure 8C–E*). At 21 dpi (but not at 10 dpi), rIL4-administered frogs possessed significantly lower skin *Bd* loads (*Figure 8F*), exhibited significantly less epidermal thickening (*Figure 8C, D, and G*) and less neutrophil infiltration (*Figure 8H–J*) than control infected animals. Conversely, subcutaneous administration of rIL4 to animals with active *Bd*-infection did not alter their fungal loads (*Figure 8—figure supplement 2B*), suggesting a critical point during which IL4 may offer protection against this fungal pathogen.

## Discussion

Amphibian extinction rates far outpace those of any other vertebrate class (*IUCN 2022, 2022*). It is now well established that chytrid fungi are major contributors to these declines, and strikingly, are considered the greatest infectious disease threat to biodiversity (*Fisher and Garner, 2020*). The development of effective mitigation strategies though is hindered by incomplete understanding of amphibian immune defenses and skin integrity. In this respect, while mast cells are recognized as key immune sentinels of tissues such as skin (*St. John et al., 2023*), relatively little is known about this cell lineage outside of mammals. Our findings provide the first in-depth functional analyses of these cells in amphibians and explore their protective mechanisms during chytrid infections. This work presents a unique perspective on the evolution of mast cell functionality and will serve as a new avenue to explore ways to counteract the amphibian declines.

Mammalian mast cells are potent producers of the pleotropic anti-inflammatory interleukin-4 (IL4) (*Babina et al., 2016*) cytokine, and our findings suggest that this is also true of frog mast cells. In mammals, IL4 also plays roles in tissue repair (*Allen, 2023*) and mucus production by intestinal goblet cells (*Redpath et al., 2015*), which aids in the 'weep and sweep' pathogen elimination and may contribute to the maintenance of commensal communities in the gut (*Drago et al., 2024*). This is consistent with our findings, which suggest that IL4 may be a central means by which frog mast cells confer protection against *Bd* by counteracting *Bd*-elicited inflammation, including minimizing neutrophil infiltration, maintaining skin integrity, and promoting mucus production by skin mucus glands. The outcomes of cytokine signaling are critically dependent on timing, location, and context (*Redpath et al., 2015*). This possibly explains why administering rIL4 prior to *Bd* challenge results in significant anti-*Bd* protection, whereas treating already infected animals with this cytokine did not reduce fungal loads.

We anticipate that in addition to IL4 production, frog mast cells mediate the observed antifungal protection through a myriad of additional mechanisms. Indeed, we observed elevated expression of antimicrobial peptide genes in the skins of mock-infected mast cell-enriched frogs 10 days after rSCF administration. The fact that we did not see the same expression patterns following *Bd* challenge could potentially reflect the highly immunomodulatory capacities of this pathogen (*Rollins-Smith and Le Sage, 2021*). Undoubtedly, frog skin-resident mast cells have evolved to contribute to and modulate skin antifungal antimicrobial peptide production just as *Bd* has more than likely coevolved to target this aspect of amphibian skin defenses.

*Bd* infections caused major reductions in bacterial taxa richness, changes in composition and substantial increases in the relative abundance of *Bd*-inhibitory bacteria early in the infection. Similar changes to microbiome structure occur during experimental *Bd* infections, such as in red-backed salamanders and mountain yellow-legged frogs (*Jani et al., 2021*; *Muletz-Wolz et al., 2019*). In turn, progressing *Bd* infections corresponded with a return to baseline levels of *Bd*-inhibitory bacteria abundance and rebounding microbial richness, albeit with dissimilar communities to those seen in control animals. These temporal changes indicate that amphibian microbiomes are dynamic, as are the effects of *Bd* infections on them. Indeed, *Bd* infections may have long-lasting impacts on amphibian microbiomes (*Jani et al., 2021*), including the possible establishment of 'ecological memory', resulting in greater adaptation to subsequent pathogen exposures (*Woodhams et al., 2023*; *Letourneau et al., 2022*). While *Bd* infections manifested in these considerable changes to frog skin microbiome structure, mast cell enrichment appeared to counteract these deleterious effects to skin microbial composition. It will thus be interesting to learn from future studies how frog mast cells impact skin microbiome ecological memory. Presumably, mast cell-mediated mucus production and preservation of skin integrity stabilizes the cutaneous microenvironment during *Bd* infections, thereby ameliorating the *Bd*-mediated microbiome changes. While this work explored the changes in established antifungal flora, we anticipate the mast cell-mediated inhibition of *Bd* may be due to additional, yet unidentified bacterial, viral, and/or fungal taxa. Intriguingly, while mammalian skin mast cell functionality depends on microbiome elicited SCF production by keratinocytes (*Wang et al., 2017*), our results indicate frog skin mast cells in turn impact skin microbiome structure and likely their function. It will be interesting to further explore the interdependent nature of amphibian skin microbiomes and resident mast cells.

In contrast to the protective effects conferred by mast cell enrichment, enrichment of neutrophils resulted in greater *Bd* burdens on frog skins. This suggests neutrophils may lead to heightened inflammation, which is not protective in the context of *Bd* infections. This is consistent with other studies

indicating that robust immune responses may be counterproductive to amphibian anti-*Bd* defenses (*Ellison et al., 2014*; *Savage et al., 2020*).

Amphibian skin is much thinner and more permeable than those of mammals (*Varga et al., 2018*), and as such, arguably represents a more penetrable barrier to pathogens. Because mammalian skin is relatively impermeable, mast cells are absent from healthy mammalian epidermises and are instead found exclusively in their dermal layers (*Cowen et al., 1979*). However, mammalian mast cells may infiltrate the skin epidermal layer during diseases such as dermatitis (*Imayama et al., 1995*). We have shown here and have observed across several classes of amphibians (*Hauser et al., 2023*) that mast cells are found in both epidermal and dermal layers. This localization has presumably evolved to support the more intimate contact between amphibian skin and their environments. Considering the importance of amphibian cutaneous integrity to their physiology (*Larsen, 2021*), it is likely that skin-resident mast cells coevolved to support skin immunity, physiology, and symbiotic microbiota. Ongoing research continues to reveal functional differences between mammalian connective tissue and mucosa-resident mast cells (*West and Bulfone-Paus, 2022*), and we suspect that there is similarly much to learn about the distinct physiological and immune roles of amphibian epidermal and dermal mast cells.

Mammalian mast cells are thought to arise from two distinct lineages. Fetal-derived progenitors seed peripheral tissues during the neonatal life (*Gentek et al., 2018*) and mature into long-lived connective tissue mast cells (CTMCs) (*Dwyer et al., 2016*). Conversely, inducible mucosal mast cells (MMCs) arise from bone marrow-derived mast cell progenitors (MCps) and are recruited to mucosal tissues in response to inflammation (*Bankova et al., 2015*; *Li et al., 2018*). The connective tissue mast cells are thought to possess significantly greater levels of heparin than bone marrow-derived mast cells (*Herrera-Heredia et al., 2022*). Interestingly, the *in vitro* frog bone marrow-derived mast cells possessed substantially lower heparin content than what we observed in the skin-resident and *Bd*-elicited mast cells. Unlike the mammalian skin, frog skin is both a connective tissue and a mucosal barrier, begging the question of how the frog skin mast cells compare to the mammalian CTMCs and MMCs. It is intriguing to consider that frog skin mast cells observed at steady state, following rSCF-enrichment, and recruited/expanded in response to *Bd*, all had substantial heparin levels. Possibly, the expansion of frog skin mast cell numbers in response to both rSCF and *Bd* is facilitated by skin-resident precursors, explaining the high heparin content observed in these populations. Of course, it is equally possible that amphibian mast cell ontogeny and heparin content are not the same as in mammals. It will be invaluable to further explore the origins, fates, and immune roles of distinct frog skin mast cell lineages and subsets.

It has become apparent that amphibian host-*Bd* interactions are highly complex and multifaceted and susceptibilities to this devastating pathogen markedly differ across amphibian species (*Scheele et al., 2019*; *Van Rooij et al., 2010*). The findings described here emphasize the importance of skin-resident mast cells for successful anti-*Bd* defenses and demonstrate that these immune sentinels are intimately linked to many aspects of frog skin physiology. Our results indicate that when mast cells are enriched, the ensuing changes in the skin allow for greater resistance to *Bd* infection. Presumably, distinct amphibian species have evolved disparate interconnections between their skin mast cells and their cutaneous defenses, as dictated by their respective physiological and environmental pressures. In turn, these species-specific differences likely dictate whether and to what extent the skin-resident mast cells of a given amphibian species recognize and appropriately respond to *Bd* infections. We postulate that such differences may contribute to the disparate susceptibilities of distinct amphibian species to pathogens like chytrid fungi. Greater understanding of these relationships across distinct amphibian species holds promise of harnessing this knowledge toward possible development of preventative and therapeutic strategies against infectious diseases like chytridiomycosis.

## Materials and methods

### Animals

Outbred tadpoles (NF stage 56–60) and 1-year-old (1.5–2″), mixed-sex *X. laevis* were purchased from *Xenopus* 1 (Dexter, MI). Purchased adult frogs were used for *in vitro* culture studies and tadpoles were reared in-house through metamorphosis and used as juveniles in skin immune composition and

*Bd*-infection studies. All animals were housed and handled under strict laboratory regulations as per GWU IACUC (approval number 15-024).

## Recombinant cytokines

The *X. laevis* rSCF, rCSF3, and rIL4 were generated as previously described for rCSF3 (*Hauser et al., 2020*). The *X. laevis* *SCF*, *IL4*, and *CSF3* sequences representing the signal peptide-cleaved transcripts were ligated into the pMIB/V5 His A insect expression vectors (Invitrogen). The resulting recombinant cytokine expression constructs or empty vectors were transfected into Sf9 insect cells (Cellfectin II, Invitrogen). Recombinant proteins contain a V5 epitope, and western blot with an anti-V5-HRP antibody (Sigma) confirmed their presence. Positive transfectants were selected using 10 µg/mL blasticidin (Gibco). Expression cultures were scaled up to 500 mL liquid cultures, grown for 5 days, pelleted by centrifugation, and the supernatants collected. Supernatants were dialyzed overnight at 4°C against 150 mM sodium phosphate, concentrated against polyethylene glycol flakes (8 kDa) at 4°C, dialyzed overnight at 4°C against 150 mM sodium phosphate, and passed through Ni-NTA agarose columns (QIAGEN). Columns were washed with 2 × 10 volumes of high-stringency wash buffer (0.5% Tween 20, 50 mM sodium phosphate, 500 mM sodium chloride, 100 mM imidazole) and 5 × 10 volumes of low-stringency wash buffer (as above but with 40 mM imidazole). Recombinant proteins were eluted with 250 mM imidazole. After recombinant protein purification, a halt protease inhibitor cocktail (containing AEBSF, aprotinin, bestatin, E-64, leupeptin, and pepstatin A; Thermo Scientific) was added. Intact recombinant protein presence was confirmed again by western blot and the protein concentrations quantified by Bradford protein assays (Bio-Rad). Protein aliquots were stored at –20°C until use.

## Bone marrow granulocyte cultures

Bone marrow isolation, culture conditions, and establishment of neutrophil cultures have been previously described (*Yaparla and Grayfer, 2018*). Briefly, adult *X. laevis* (approximately 1 year old) were euthanized in 5% tricaine mesylate followed by cervical dislocation. Femurs were removed and washed in ice-cold Amphibian-PBS (A-PBS) in sterile conditions. Each femur was flushed with 5 mL of A-PBS. Red blood cells were removed from culture via a differential gradient generated with 51% Percoll (Invitrogen). Bone marrow cell counts were generated using trypan blue exclusion and cells were seeded at a density of $10^4$ cells/well for gene expression experiments, $5 \times 10^4$ cells/well for histology analyses, and $10^5$ cells/well for electron microscopy analyses.

Mast cell cultures were generated according to protocols adapted from *Koubourli et al., 2018* and *Meurer et al., 2016*. Isolated bone marrow cells were treated with 250 ng/mL of rSCF on days 0, 4, and 7, and collected for further analysis on day 9. Cell cultures were maintained at 27°C with 5% $CO_2$ in amphibian medium supplemented with 10% fetal bovine serum and 0.25% *X. laevis* serum. Neutrophil-like granulocytes were generated as above but with 250 ng/mL of rCSF3 on days 0 and 3, and collected for further analysis on day 5. Cell cultures were maintained at 27°C with 5% $CO_2$ in amphibian serum-free medium supplemented with 10% fetal bovine serum, 0.25% *X. laevis* serum, 10 µg/mL gentamicin (Thermo Fisher Scientific, Waltham, MA), 100 U/mL penicillin, and 100 µg/mL streptomycin (Gibco, Thermo Fisher Scientific).

## Enrichment of skin granulocyte subsets

Animals were subcutaneously injected between the skin and muscle layers with 5 µg/animal of rSCF, rCSF3, or r-ctrl in 10 µL of saline using finely pulled glass needles. Optimal time course and dose for *in vivo* mast cell and neutrophil enrichment were determined during preliminary experiments.

## Recombinant interleukin-4 treatment

The capacity of the recombinant interleukin-4 (rIL4) to induce expression of genes associated with mammalian IL4 responses were assessed by injecting frogs (N = 6 per treatment group) subcutaneously with rIL4 (5 µg/animal) or r-ctrl in 10 µL of saline. After 6 hr, animals were sacrificed, and skins were isolated for gene expression analyses.

## *Bd* stocks and fungal challenge

*Bd* isolate JEL 197 was grown in 1% tryptone broth or on 1% tryptone agar plates (Difco Laboratories, Detroit, MI) supplemented with 100 U/mL penicillin and 100 µg/mL streptomycin (Gibco) at 19°C.

*In vitro Bd* killing was evaluated by incubating live *Bd* (maturing zoosporangia) with mast cells or neutrophils at ratios of 5:1 or 1:1 *Bd* cells per granulocyte. Cells were incubated at 27°C for 3 days before fungal loads were analyzed by absolute qPCR. Experimental groups were compared to pathogen DNA amounts derived from equal quantities of live *Bd* plated alone in otherwise identical conditions.

For *in vivo* infection studies, zoospores were harvested by flooding confluent tryptone agar plates with 2 mL sterile A-PBS for 10 min. Twelve hr post rSCF, rCSF3, rIL4, or r-ctrl injection, animals were infected with $10^7$ zoospores or mock-infected in 100 mL of water. After 3 hr, 400 mL of water was added to each tank. Skins were collected for histology and gene expression analyses on 1, 10, and 21 dpi, with 10 and 21 dpi representing intermediate and later time points of infection, respectively.

To examine the effect of rIL4 on *Bd* loads, frogs were infected with *Bd* by water bath ($10^7$ zoospores, as described above) and 1 day later injected subcutaneously, dorsally with rIL4 (5 µg/animal) or r-ctrl in 10 µL of saline. After an additional 9 days of infection, animals were sacrificed and their dorsal skin *Bd* loads examined.

## Analyses of mucus *Bd*-killing capacities

Mucosomes were collected from mast cell- or vector-enriched *X. laevis* that were either mock- or *Bd*-infected for 10 or 21 days. To this end, individual *X. laevis* were soaked in a 5 mL water bath for 1 hr. Each water sample was then lyophilized, reconstituted with 500 µL of molecular grade water, and passed through a sterile cell strainer to remove large debris.

*Bd* was seeded in opaque white 96-well plates (20,000 zoospores in 50 µL of tryptone broth/well). Next, 50 µL of mucosome solution was added to each well (100 µL total well volume) in three replicate wells per individual *X. laevis* mucosome. Mucosomes, tryptone broth, and water were each plated alone as controls. Plates were sealed with parafilm and incubated at 19°C for 16 hr with gentle mixing (20 rpm).

Zoospore viability was determined with the CellTiter-Glo 2.0 Cell Viability assay kit (Promega) according to the manufacturer's instructions and using a SpectraMax plate reader (Molecular Devices, San Jose, CA). Luminescence readings were fitted to a standard curve (descending proportions of heat-killed zoospores to viable zoospores) to calculate the number of viable zoospores in each well. Zoospores were heated-killed at 65°C for 15 min.

## Histology

Leukocyte cytology and cutaneous SE staining has been described (*Hauser et al., 2020*). An Alcian Blue/PAS staining kit (Newcomer Supply, Middleton, WI) was used to quantify mucin content. Paraffin-embedded tissue sections (5 µm) were deparaffinized, rehydrated through A-PBS, and stained with Naphthol AS-D Chloroacetate (specific esterase; Sigma) or Alcian Blue/PAS (Newcomer Supply) according to the manufacturer's instructions and optimized for *Xenopus* skin tissues. Cells collected from *in vitro* cultures were cytocentrifuged onto glass microscope slides (VWR). Cells were stained immediately with Giemsa (Sigma) for 7 min or fixed with 10% neutral-buffered formalin for 30 min and stained with specific esterase according to the manufacturer's instructions. Slides stained with Alcian Blue/PAS (Newcomer Supply) were used to quantify mucin content from *in vivo* experiments. Images were taken using identical microscope settings under ×20 magnification. Images were converted to 8-bit in Fiji by ImageJ and threshold adjusted such that positive staining for mucus was captured within the mucus glands (threshold held constant across images). The percentage of each mucus gland positively stained and the average percent-positive per field of view were subsequently calculated. Positive staining of both acidic and neutral mucins was included in analyses. ImageJ was also used for epidermal thickness analyses using scale bars in the images to calibrate and measure epidermal thickness. All slides were imaged with a Leica DMi8 Inverted Fluorescent Microscope with all mucus glands assessed for each respective frog skin section (Leica Microsystems, Davie, FL).

An RNAScope ISH Kit (ACD Bio) and a *X. laevis* myeloperoxidase (mpo)-specific probe (ACD Bio) were used according to the manufacturer's instructions to visualize mpo-positive neutrophils in frog skin tissues.

Toward avidin staining, skin tissues were fixed in 4% paraformaldehyde, washed with saline, cryo-protected in 15% then 30% sucrose, flash-frozen in optimal cutting temperature (OCT) compound (Fisher), and cryo-sectioned onto microscope slides (Fisher). Frozen sections were stained with Texas

red-conjugated avidin (Thermo Fisher) and DAPI (Thermo Fisher) and glass cover slips mounted with Prolong Antifade mounting media (Thermo Fisher). Tissues were imaged using a Zeiss LSCM-800 confocal microscope. For each slide, 15 fields of view were enumerated at ×20 (Plan-Apochromat 20×/0.75) objective. Images were inversed in ImageJ to improve contrast and resolution of heparin-positive skin mast cells.

## Electron microscopy

Processing and imaging of cells for TEM and SEM was conducted at the GWU Nanofabrication and Imaging Center (GWNIC). For TEM, cells were fixed as monolayers on six-well plates with 2.5% glutaraldehyde and 1% paraformaldehyde in 0.1 M sodium cacodylate buffer for 1 hr. Cells were treated with 1% osmium tetroxide in 0.1 M sodium cacodylate buffer for 1 hr. Following washes, cells were *en bloc* stained with 1% uranyl acetate in water overnight at 4°C. Samples were dehydrated through an ethanol series and embedded in epoxy resin using LX112. Inverted BEEM capsules were placed into each tissue culture well to create *on face* blockfaces for sectioning. Resin was cured for 48 hr at 60°C. The 95 nm sections were post-stained with 1% aqueous uranyl acetate and Reynold's lead citrate. All imaging was performed at 80 kV in a Talos 200X transmission electron microscope (Thermo Fisher Scientific, Hillsboro, OR).

For SEM, cells were fixed with 2.5% glutaraldehyde/1% paraformaldehyde in sodium cacodylate buffer, followed by 1% $OsO_4$, then dehydrated through an ethyl alcohol series. Coverslips were critical point dried and coated with 2 nm iridium. Cells were imaged using a Teneo Scanning Electron Microscope (Thermo Fisher Scientific).

## Analyses of immune gene expression and *Bd* skin loads

These analyses have been described (*Hauser et al., 2020*). Cells and tissues were homogenized in Trizol reagent, flash-frozen on dry ice, and stored at –80°C until RNA and DNA isolation. RNA isolation was performed using Trizol according to the manufacturer's directions. RNA-Seq is described in detail below. For qRT-PCR gene expression analysis, total RNA (500 ng/sample) was reverse-transcribed into cDNA using cDNA qscript supermix (Quantabio, Beverly, MA). Following RNA extraction, back extraction buffer (4 M guanidinethiocyanate, 50 mM sodium citrate, 1 M Tris pH 8.0) was mixed with the remaining Trizol layer and centrifuged to isolate the DNA-containing aqueous phase. DNA was precipitated overnight with isopropanol, pelleted by centrifugation, washed with 80% ethanol, and resuspended in TE buffer (10 mM Tris pH 8.0, 1 mM EDTA). DNA was purified by phenol:chloroform extraction and resuspended in molecular grade water (VWR).

Quantitative gene expression analyses for both *Bd* and *X. laevis* cells and tissues were performed using the CFX96 Real-Time System (Bio-Rad Laboratories, Hercules, CA) and iTaq Universal SYBR Green Supermix (Bio-Rad Laboratories). The Bio-Rad CFX Manager software (SDS) was employed for all expression analysis. All expression analyses were conducted using the ΔΔCt method relative to the *gapdh* endogenous control gene for *X. laevis*. Fungal load quantification was assessed by absolute qPCR. Isolated *Bd* DNA (JEL 197 isolate) was serially diluted and used as the standard curve. Primers were designed and validated against the *Bd* ribosomal RNA internal transcribed spacer 1 (ITS1). The primers used are listed in *Supplementary file 1*.

## RNA sequencing

For transcriptomic profiling, bone marrow-derived neutrophil and mast cell cultures were generated as described above and FACS-sorted according to pre-established size and internal complexity parameters to isolate the respective subsets for further analyses. Sorted cells were immediately processed to extract and purify RNA. Flash-frozen samples were sent to Azenta Life Sciences for all library preparation, RNA sequencing, and analyses. In short, polyadenylated RNA was isolated using Oligo dT beads. Enriched mRNAs were then fragmented for first- and second-strand cDNA synthesis. cDNA fragments were end repaired, 5′ phosphorylated, and dA-tailed. Fragments were then ligated to universal adaptors and PCR-amplified. 150 bp paired-end sequencing was performed on an Illumina HiSeq platform.

FastQC was used to evaluate raw data quality. Adaptors sequences and poor-quality nucleotides were removed from reads using Trimmomatic v.0.36. The STAR aligner v.2.55.2b was used to map these reads to the *Xenopus*_laevis_9_2 reference genome from ENSEMBL. To determine differential

gene expression, featureCount (Subread package v.1.5.2) was first used to count unique gene hits, which were then used with DESeq2 to calculate absolute $\log_2$ fold change.

## Skin microbiome analyses

Towards microbiome studies, frogs were housed individually (N = 6/treatment group). At indicated times, frogs were gently rinsed with sterile deionized water to remove transient microbes and gently swabbed 20 times, dorsally. Genomic DNA was extracted from swabs using a PowerSoil Pro kit on a Qiacube HT (QIAGEN, MD). One-step PCR library prep and dual-index paired-end Illumina sequencing was used to sequence the skin microbiome of individual frogs. An ~380 base pair region in the V3-V5 region of the 16S rRNA gene using the universal primers 515F-Y (GTGYCAGCMGCCGCGGTAA) and 939R (CTTGTGCGGGCCCCCGTCAATTC) was used for amplification. Negative and positive controls (ZymoBIOMICS D6300 & D6305, Zymo, CA) were included in each round of extraction and PCR. Reactions were done in duplicate for each sample, pooled, cleaned with in-house Speed-beads (in a PEG/NaCl buffer), quantified with a Qubit4 (Invitrogen, MA) and pooled into a final library in equimolar proportion. The pooled library was sequenced on two Illumina MiSeq runs (v3 chemistry: 2 × 300 bp kit) at the Center for Conservation Genomics, Smithsonian National Zoo & Conservation Biology Institute.

All analyses were performed in the R environment version 4.0.3 (R Core Team, 2020). Demultiplexed reads were imported from Basespace into R environment for sequence processing. Package 'dada2' (*Callahan et al., 2016*) was used to perform quality filtering using their standard filtering parameters (i.e., maxEE = 2), which collapsed high-quality reads into ASV and removed chimeras. Bacterial taxonomy was assigned using Silva version 138.1. The R package 'phyloseq' (*McMurdie and Holmes, 2013*) was used to import and merge the final ASV table, taxonomy table, and metadata to create a phyloseq object to perform further analyses. Sequences classified as cyanobacteria/chloroplast and those unclassified at kingdom were removed. Singletons were filtered out (i.e., ASVs with only one sequence read in one individual). The R package 'decontam' (*Davis et al., 2018*) was used to remove potential contaminants using the method 'combined'. The ZymoBIOMICS microbial community standards (positive controls) were analyzed, and we found genera in similar relative abundances as described by Zymo.

To determine how *Bd* and mast cell treatments impacted skin microbiomes, the microbiome structure was examined. The components of microbiome structure were ASV richness (measured as *Bd*-inhibitory ASV richness and total ASV richness), microbial composition (measured by Jaccard and Bray–Curtis distances), and sequence abundance of *Bd*-inhibitory ASVs (measured as individual *Bd*-inhibitory ASV sequence counts and total relative abundance of *Bd*-inhibitory ASVs). To characterize variation in microbiome structure, mast cell (mast cell normal and mast cell+), *Bd* (*Bd-* and *Bd*+), and their interaction as explanatory variables at two time points days 10 and 21 post *Bd* infection were included. For this characterization, log-transformed ASV richness in ANOVAs, microbial composition measures in PERMANOVAs and log-transformed raw sequence counts in ANOVAs (with post hoc corrections for multiple comparisons) were used. For identification of *Bd*-inhibitory ASVs, methods as described in *Jiménez et al., 2022* were followed.

Days 10 and 21 post *Bd* infection were chose for these analyses since they represent an intermediate and a later timepoint of infection.

## Statistical analyses

Differences in transcript expression were calculated with one-way or multiway ANOVAs followed by Tukey post hoc tests. Student's *t*-tests were used to determine differences in *Bd* loads between treatments only. Statistical differences in mucin content and mucosome *Bd*-killing were assessed with the two-way ANOVA calculator available online through Statistics Kingdom. For RNA sequencing, p-values were calculated with the Wald test and were adjusted using the Benjamini–Hochberg procedure.

## Acknowledgements

We thank Dr. Cynthia S Dowd and Ben England (Chemistry Department, GWU) for their help with *X. laevis* mucus lyophilization. We thank Dr. Christine Brantner (Nanofabrication and Imaging Center, GWU) for assistance with TEM and SEM imaging. We thank Dr. Alexander Jeremic (Biology Department, GWU) for access to and use of his confocal microscope. We thank Dr. Gregory Cresswell (Flow

Cytometry Core, GWU) for his help with flow cytometry and cell sorting. We thank Dr. Karyn Jourdeuil (NIH) for assistance with tissue preparation and cryo-sectioning. KAH, MRHH, NR, NK, AY, MZ, and LG thank the GWU Biology Department. KAH, MRHH, and AY were supported by Wilbur V Harlan summer graduate research fellowships. We thank the three anonymous reviewers whose insightful comments and suggestions helped to improve the scope and breadth of this article. This work was supported by National Science Foundation grant 2131061 (LG, CRMW), National Science Foundation grant 2147466 (LG), and National Science Foundation grant 2147467 (LAR-S)

## Additional information

### Funding

| Funder | Grant reference number | Author |
|---|---|---|
| National Science Foundation | 2131061 | Carly R Muletz-Wolz Leon Grayfer |
| National Science Foundation | 2147466 | Leon Grayfer |
| National Science Foundation | 2147467 | Louise A Rollins-Smith |

The funders had no role in study design, data collection and interpretation, or the decision to submit the work for publication.

### Author contributions

Kelsey A Hauser, Conceptualization, Data curation, Formal analysis, Investigation, Methodology, Writing – original draft, Writing – review and editing; Christina N Garvey, Muhammad RH Hossainey, Data curation, Investigation, Visualization, Methodology; Ryley S Crow, Conceptualization, Data curation, Validation, Investigation, Visualization, Methodology; Dustin T Howard, Netra Ranganathan, Namarta Kalia, Mira Zelle, Data curation, Investigation, Methodology; Lindsey K Gentry, Data curation, Methodology; Amulya Yaparla, Data curation, Formal analysis, Investigation, Methodology; Elizabeth J Jones, Anju N Duttargi, Investigation, Methodology; Louise A Rollins-Smith, Conceptualization, Data curation, Formal analysis, Funding acquisition, Investigation, Methodology, Writing – original draft, Writing – review and editing; Carly R Muletz-Wolz, Conceptualization, Data curation, Formal analysis, Funding acquisition, Validation, Investigation, Visualization, Methodology, Writing – original draft, Writing – review and editing; Leon Grayfer, Conceptualization, Resources, Data curation, Formal analysis, Supervision, Funding acquisition, Validation, Investigation, Visualization, Methodology, Writing – original draft, Project administration, Writing – review and editing

### Author ORCIDs

Muhammad RH Hossainey ⬡ https://orcid.org/0000-0003-4780-7509
Leon Grayfer ⬡ https://orcid.org/0000-0002-1144-4884

### Ethics

All animals were housed and handled under strict laboratory regulations as per GWU IACUC (Approval number 15-024).

Reviewer #1 (Public Review): https://doi.org/10.7554/eLife.92168.3.sa1
Reviewer #2 (Public Review): https://doi.org/10.7554/eLife.92168.3.sa2
Author response https://doi.org/10.7554/eLife.92168.3.sa3

## Additional files

### Supplementary files
• MDAR checklist
• Supplementary file 1. List of primer sequences.

## Data availability

Illumina 16S rRNA sequence data and associated metadata have been deposited in the National Center for Biotechnology Information Sequence Read Archive (https://www.ncbi.nlm.nih.gov/sra) under BioProject ID: PRJNA1061169. Microbiome files, analysis and R code are available on Github: https://github.com/CarlyMuletzWolz/MastCellsGuardians, copy archived at *Muletz-Wolz, 2024*. The RNAseq comparisons of bone marrow-derived mast cell and neutrophil cultures and comparisons of control and mast cell-enriched, Bd-infected skins have been deposited to Dryad: https://doi.org/10.5061/dryad.bcc2fqznc.

The following datasets were generated:

| Author(s) | Year | Dataset title | Dataset URL | Database and Identifier |
|---|---|---|---|---|
| Hauser KA, Garvey CN, Crow RS, Hossainey MRH, Howard DT, Ranganathan N, Gentry LK, Yaparla A, Kalia N, Zelle M, Jones EJ, Duttargi AN, Rollins-Smith LA, Muletz-Wolz CR, Grayfer L | 2024 | Data from: Analyses *Xenopus laevis* mast cells, neutrophils, and mast cell-enriched, chytrid infected skin | https://doi.org/10.5061/dryad.bcc2fqznc | Dryad Digital Repository, 10.5061/dryad.bcc2fqznc |
| Hauser KA, Gentry LK, Grayfer L, Muletz-Wolz CR | 2024 | Amphibian mast cells: barriers to deadly chytrid fungus infections | https://www.ncbi.nlm.nih.gov/bioproject/PRJNA1061169/ | NCBI BioProject, PRJNA1061169 |

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
