## [Editor Report · eLife assessment]

This **important** study reveals the role of skin-resident mast cells in amphibians in mediating antimicrobial responses. The data are **compelling** and highlight species-specific biology that can cross-inform human mast cell biology in a species that does not rely on IgE as a primary mechanism for antimicrobial skin responses.

---

## [Referee Report · Reviewer #1 (Public Review)]

Summary:

The global decline of amphibians is primarily attributed to deadly disease outbreaks caused by the chytrid fungus, Batrachochytrium dendrobatidis (Bd). It is unclear whether and how skin-resident immune cells defend against Bd. Although it is well known that mammalian mast cells are crucial immune sentinels in the skin and play a pivotal role in immune recognition of pathogens and orchestrating subsequent immune responses, the roles of amphibian mast cells during Bd infections is largely unknown. The current study developed a novel way to enrich *X. laevis* skin mast cells by injecting the skin with recombinant stem cell factor (SCF), a KIT ligand required for mast cell differentiation and survival. The investigators found an enrichment of skin mast cells provides *X. laevis* substantial protection against Bd and mitigates the inflammation-related skin damage resulting from Bd infection. Additionally, the augmentation of mast cells leads to increased mucin content within cutaneous mucus glands and shields frogs from the alterations to their skin microbiomes caused by Bd.

Strengths:

This study underscores the significance of amphibian skin-resident immune cells in defenses against Bd and introduces a novel approach to examining interactions between amphibian hosts and fungal pathogens.

Weaknesses:

The main weakness of the study is lack of functional analysis of *X. laevis* mast cells. Upon activation, mast cells have the characteristic feature of degranulation to release histamine, serotonin, proteases, cytokines, and chemokines, etc. The study should determine whether *X. laevis* mast cells can be degranulated by two commonly used mast cell activators IgE and compound 48/80 for IgE-dependent and independent pathway. This can be easily done *in vitro*. It is also important to assess whether *in vivo* these mast cells are degranulated upon Bd infection using avidin staining to visualize vesicle releases from mast cells. Figure 3 only showed rSCF injection caused an increase in mast cells in naïve skin. They need to present whether Bd infection can induce mast cell increase and rSCF injection under Bd infection causes a mast cell increase in the skin. In addition, it is unclear how the enrichment of mast cells provides the protection against Bd infection and alternations to skin microbiomes after infection. It is important to determine whether skin mast cell release any contents mentioned above.

---

## [Referee Report · Reviewer #2 (Public Review)]

Summary:

In this study, Hauser et al investigate the role of amphibian (*Xenopus laevis*) mast cells in cutaneous immune responses to the ecologically important pathogen Batrachochytrium dendrobatidis (Bd) using novel methods of *in vitro* differentiation of bone marrow-derived mast cells and *in vivo* expansion of skin mast cell populations. They find that bone marrow-derived myeloid precursors cultured in the presence of recombinant *X. laevis* Stem Cell Factor (rSCF) differentiate into cells that display hallmark characteristics of mast cells. They inject their novel (r)SCF reagent in the skin of *X. laevis* and find that this stimulates expansion of cutaneous mast cell populations *in vivo*. They then apply this model of cutaneous mast cell expansion in the setting of Bd infection and find that mast cell expansion attenuates skin burden of Bd zoospores and pathologic features including epithelial thickness and improves protective mucus production and transcriptional markers of barrier function. Utilizing their prior expertise with expanding neutrophil populations in *X. laevis*, the authors compare mast cell expansion using (r)SCF to neutrophil expansion using recombinant colony stimulating factor 3 (rCSF3) and find that neutrophil expansion in Bd infection leads to greater burden of zoospores and worse skin pathology. Combining these two observations, they demonstrate that mast cell expansion using rSCF attenuates cutaneous neutrophilic infiltration. They further show that mast cell expansion correlates to cutaneous IL-4 expression, and that treatment with exogenous rIL-4 reduces neutrophilic infiltration and restores markers of epithelial health, offering a mechanism by which mast cell expansion protects from Bd infection.

Strengths:

The authors report a novel method of expanding amphibian mast cells utilizing their custom-made rSCF reagent. They rigorously characterize expanded mast cells *in vitro* and *in vivo* using histologic, morphologic, transcriptional, and functional assays. This establishes solid footing with which to then study the role of rSCF-stimulated mast cell expansion in the Bd infection model. This appears to be the first demonstration of exogenous use of rSCF in amphibians to expand mast cell populations and may set a foundation for future mechanistic studies of mast cells in the *X. laevis* model organism. Building on prior work, they are able to contrast mast cell expansion with their neutrophil expansion model, allowing them to infer a mechanistic link between mast cell expansion and IL-4 production and subsequent suppression of neutrophil infiltration and cutaneous dysbiosis.

Weaknesses:

The main weaknesses derive from technical limitations inherent to the *Xenopus* model at this time. For example, in mice a mechanistic study would be expected to use IL-4 knockouts, preferably mast cell-specific, to prove the link between mast cell expansion and IL-4 production being necessary and sufficient to suppress neutrophils. However, the novel reagents in this manuscript present a compelling technical advance and a step forward in the tools available to study amphibian biology.

In addition to their discussion, one open question from the revised manuscript is how a single treatment with rSCF leads to a peak in mast cell numbers and then decline to baseline in mock-infected frogs, while Bd infection either sustains rSCF-boosted mast cells or leads to steady mast cell increase over time in control-treated frogs. Whether this is mediated by endogenous SCF or some other factor remains unexplored.

---

## [Author Response]

Author response:

The following is the authors’ response to the current reviews.

**Reviewer #1(Public Review):**
Summary: The global decline of amphibians is primarily attributed to deadly disease outbreaks caused by the chytrid fungus, Batrachochytrium dendrobatidis (Bd). It is unclear whether and how skin-resident immune cells defend against Bd. Although it is well known that mammalian mast cells are crucial immune sentinels in the skin and play a pivotal role in immune recognition of pathogens and orchestrating subsequent immune responses, the roles of amphibian mast cells during Bd infections is largely unknown. The current study developed a novel way to enrich *X. laevis* skin mast cells by injecting the skin with recombinant stem cell factor (SCF), a KIT ligand required for mast cell differentiation and survival. The investigators found an enrichment of skin mast cells provides *X. laevis* substantial protection against Bd and mitigates the inflammation-related skin damage resulting from Bd infection. Additionally, the augmentation of mast cells leads to increased mucin content within cutaneous mucus glands and shields frogs from the alterations to their skin microbiomes caused by Bd.Strengths: This study underscores the significance of amphibian skin-resident immune cells in defenses against Bd and introduces a novel approach to examining interactions between amphibian hosts and fungal pathogens.

We thank the reviewer for recognizing the significance and the novelty of our work.

Weaknesses: The main weakness of the study is lack of functional analysis of *X. laevis* mast cells. Upon activation, mast cells have the characteristic feature of degranulation to release histamine, serotonin, proteases, cytokines, and chemokines, etc. The study should determine whether *X. laevis* mast cells can be degranulated by two commonly used mast cell activators IgE and compound 48/80 for IgE-dependent and independent pathway. This can be easily done *in vitro*. It is also important to assess whether *in vivo* these mast cells are degranulated upon Bd infection using avidin staining to visualize vesicle releases from mast cells. Figure 3 only showed rSCF injection caused an increase in mast cells in naïve skin. They need to present whether Bd infection can induce mast cell increase and rSCF injection under Bd infection causes a mast cell increase in the skin. In addition, it is unclear how the enrichment of mast cells provides the protection against Bd infection and alternations to skin microbiomes after infection. It is important to determine whether skin mast cell release any contents mentioned above.

We would like to thank the reviewer for taking the time to review our work and for providing us with valuable feedback.

Please note, that as indicated in our previous rebuttal to reviewers, amphibians do not possess the IgE antibody isotype (Flajnik, 2018).

To our knowledge, there are no published works describing the approaches used in studying mammalian mast cell degranulation towards examining amphibian mast cells. While there are commercially available kits and reagents for examining mammalian mast cell granule content, most of these do not cross-react with amphibian counterparts. This is especially true of cytokines and chemokines, which diverged quickly with evolution and thus do not share substantial protein sequence identity across species as diverged as frogs and mammals. We would also like to highlight the fact that several studies suggest that amphibian mast cells lack histamine (Mulero et al., 2007; Reite, 1965; Reite, 1972; Takaya et al., 1967) and serotonin (Mulero et al., 2007; Galli, 1990). While following up on these findings would be possible, we would like to respectfully emphasize that adopting approaches used in mammalian research to comparative immunology work is not always straightforward.

As we highlight in our manuscript, frog mast cells upregulate their expression of interleukin-4 (IL4), a hallmark cytokine associated with mammalian mast cells (Babina et al., 2016). The additional findings presented in our revised manuscript indicate that mast cells respond to *Bd* by upregulating IL4 expression *in vitro* and *in vivo*. Together, this suggests that IL4 may be a central means by which frog mast cells confer protection against *Bd*, by counteracting *Bd*-elicited inflammation, including minimizing neutrophil infiltration, maintaining skin integrity, and promoting cutaneous mucus production. Please find that these additional results are presented in Figure 8 and are described in the results and discussion sections of our revised manuscript.

Our attempts to elicit degranulation of frog mast cells using compound 48/80 have so far not been successful. This may reflect technical issues with assays optimized for mammalian mast cells or biological difference between frog and mammalian mast cells, such as species differences in mas-related G-protein coupled receptors, through which compound 48/80 acts (Hermans et al., 2021). We will continue to explore means to study frog mast cell degranulation both *in vitro* and *in vivo* but also respectfully point out that while degranulation is a feature commonly associated with mammalian mast cells, this is not the only means by which the mammalian mast cells confer their immunological effects. Indeed, our studies suggest that frog mast cell IL4 production may be a key means by which these cells offer anti-*Bd* protection.

Please note that we successfully adopted an avidin staining approach to visualize mast cell heparin content *in vitro* and to evaluate cutaneous mast cell numbers *in vivo* in control and mast cell-enriched, mock- and *Bd*-infected animals. This additional work is depicted in Figure 4 and addressed in the results and discussion sections of our revised manuscript.

**Reviewer #2 (Public Review):**
Summary: In this study, Hauser et al investigate the role of amphibian (*Xenopus laevis*) mast cells in cutaneous immune responses to the ecologically important pathogen Batrachochytrium dendrobatidis (Bd) using novel methods of *in vitro* differentiation of bone marrow-derived mast cells and *in vivo* expansion of skin mast cell populations. They find that bone marrow-derived myeloid precursors cultured in the presence of recombinant *X. laevis* Stem Cell Factor (rSCF) differentiate into cells that display hallmark characteristics of mast cells. They inject their novel (r)SCF reagent in the skin of *X. laevis* and find that this stimulates expansion of cutaneous mast cell populations *in vivo*. They then apply this model of cutaneous mast cell expansion in the setting of Bd infection and find that mast cell expansion attenuates skin burden of Bd zoospores and pathologic features including epithelial thickness and improves protective mucus production and transcriptional markers of barrier function. Utilizing their prior expertise with expanding neutrophil populations in *X. laevis*, the authors compare mast cell expansion using (r)SCF to neutrophil expansion using recombinant colony stimulating factor 3 (rCSF3) and find that neutrophil expansion in Bd infection leads to greater burden of zoospores and worse skin pathology. Combining these two observations, they demonstrate that mast cell expansion using rSCF attenuates cutaneous neutrophilic infiltration. They further show that mast cell expansion correlates to cutaneous IL-4 expression, and that treatment with exogenous rIL-4 reduces neutrophilic infiltration and restores markers of epithelial health, offering a mechanism by which mast cell expansion protects from Bd infection.Strengths: The authors report a novel method of expanding amphibian mast cells utilizing their custom-made rSCF reagent. They rigorously characterize expanded mast cells *in vitro* and *in vivo* using histologic, morphologic, transcriptional, and functional assays. This establishes solid footing with which to then study the role of rSCF-stimulated mast cell expansion in the Bd infection model. This appears to be the first demonstration of exogenous use of rSCF in amphibians to expand mast cell populations and may set a foundation for future mechanistic studies of mast cells in the *X. laevis* model organism. Building on prior work, they are able to contrast mast cell expansion with their neutrophil expansion model, allowing them to infer a mechanistic link between mast cell expansion and IL-4 production and subsequent suppression of neutrophil infiltration and cutaneous dysbiosis.

We thank the reviewer for recognizing the rigorousness and utility of the studies presented in our manuscript.

Weaknesses: The main weaknesses derive from technical limitations inherent to the *Xenopus* model at this time. For example, in mice a mechanistic study would be expected to use IL-4 knockouts, preferably mast cell-specific, to prove the link between mast cell expansion and IL-4 production being necessary and sufficient to suppress neutrophils. However, the novel reagents in this manuscript present a compelling technical advance and a step forward in the tools available to study amphibian biology.

We agree with the reviewer that an IL4 knock-out animal model would be a great way to support our findings. Unfortunately, working with a non-mammalian model such as *X. laevis* poses limitations that include lack of knock-out lines for immunology research. Moreover, as mentioned in our manuscript, we do not believe that IL4 is the sole mast cell-produced component responsible for the conferred antifungal protection. We thank the reviewer for acknowledging the limitations of our model system and recognizing the novelty, technical advances, and merits of the work presented in our manuscript.

In addition to their discussion, one open question from the revised manuscript is how a single treatment with rSCF leads to a peak in mast cell numbers and then decline to baseline in mock-infected frogs, while Bd infection either sustains rSCF-boosted mast cells or leads to steady mast cell increase over time in control-treated frogs. Whether this is mediated by endogenous SCF or some other factor remains unexplored.

This is an interesting question that we hope to explore in future studies. We did not see significant differences in skin SCF gene expression at 21 days post *Bd* infection. This does not rule out the possibility that the observed *Bd*-mediated effects to frog skin mast cell composition are not due to changes in skin SCF gene expression at earlier infection times, alone or in combination with other host or pathogen derived factors. We know that other factors are responsible for homing/retention of antimicrobial and immunosuppressive granulocyte subsets within frog skin (Hauser et al., 2020) and we postulate that some of these may be distinct mast cell types. Additionally, *Bd* is known to produce a myriad of immunomodulatory factors (Rollins-Smith and Le Sage, 2021), which may well also directly affect frog skin mast cell composition. Mammalian mast cells are heterogenous and are homed or recruited into tissues by an extensive array of host as well as microbiome-derived components (Halova et al., 2012; West and Bulfone-Paus, 2022). Undoubtedly, the frog skin mast cell composition is likewise complex, dynamic, and contingent on a plethora of host, cutaneous microbial flora- and in this case also *Bd*-produced factors.

**References**

Babina M, Guhl S, Artuc M, Zuberbier T. 2016. IL-4 and human skin mast cells revisited: reinforcement of a pro-allergic phenotype upon prolonged exposure. *Arch Dermatol Res*
**308**:665.

Flajnik MF. 2018. A cold-blooded view of adaptive immunity. *Nat Rev Immunol*
**18**:438.

Galli SJ. 1990. New insights into "the riddle of the mast cells": microenvironmental regulation of mast cell development and phenotypic heterogeneity. *Lab Invest*
**62**:5.

Halova I, Draberova L, Draber P. 2012. Mast cell chemotaxis - chemoattractants and signaling pathways. *Front Immunol*
**3**:119.

Hauser K, Popovic M, Koubourli D, Reeves P, Batheja A, Webb R, Forzan M, Grayfer L. 2020. Discovery of granulocyte-lineage cells in the skin of the amphibian *Xenopus laevis*. *FACETS*
**5**:571.

Hermans MAW, van Stigt AC, van de Meerendonk S, Schrijver B, van Daele PLA, van Hagen PM, van Splunter M, Dik WA. 2021. Human Mast Cell Line HMC1 Expresses Functional Mas-Related G-Protein Coupled Receptor 2. *Front Immunol*
**12**:625284.

Mulero I, Sepulcre MP, Meseguer J, Garcia-Ayala A, Mulero V. 2007. Histamine is stored in mast cells of most evolutionarily advanced fish and regulates the fish inflammatory response. *Proc Natl Acad Sci U S A*
**104**:19434.

Reite OB. 1965. A phylogenetical approach to the functional significance of tissue mast cell histamine. *Nature*
**206**:1334.

Reite OB. 1972. Comparative physiology of histamine. *Physiol Rev*
**52**:778-819.

Rollins-Smith LA, Le Sage EH. 2021. Batrachochytrium fungi: stealth invaders in amphibian skin. *Curr Opin Microbiol*
**61**:124.

Takaya K, Fujita T, Endo K. 1967. Mast cells free of histamine in Rana catasbiana. *Nature*
**215**:776-7.

West PW, Bulfone-Paus S. 2022. Mast cell tissue heterogeneity and specificity of immune cell recruitment. *Front Immunol*
**13**:932090.

Babina M, Guhl S, Artuc M, Zuberbier T. 2016. IL-4 and human skin mast cells revisited: reinforcement of a pro-allergic phenotype upon prolonged exposure. A*rchives of dermatological research*
**308**:665.

Buchan KD, Prajsnar TK, Ogryzko NV, de Jong NWM, van Gent M, Kolata J, Foster SJ, van Strijp JAG, Renshaw SA. 2019. A transgenic zebrafish line for *in vivo* visualisation of neutrophil myeloperoxidase. *PLoS One*
**14**:e0215592.

Flajnik MF. 2018. A cold-blooded view of adaptive immunity. *Nat Rev Immunol*
**18**:438-453.

Galli SJ. 1990. New insights into "the riddle of the mast cells": microenvironmental regulation of mast cell development and phenotypic heterogeneity. *Lab Invest*
**62**:5.

Hermans MAW, van Stigt AC, van de Meerendonk S, Schrijver B, van Daele PLA, van Hagen PM, van Splunter M, Dik WA. 2021. Human Mast Cell Line HMC1 Expresses Functional Mas-Related G-Protein Coupled Receptor 2. *Front Immunol*
**12**:625284.

Jamur MC, Moreno AN, Mello LF, Souza Junior DA, Campos MR, Pastor MV, Grodzki AC, Silva DC, Oliver C. 2010. Mast cell repopulation of the peritoneal cavity: contribution of mast cell progenitors versus bone marrow derived committed mast cell precursors. B*MC Immunol*
**11**:32.

Mulero I, Sepulcre MP, Meseguer J, Garcia-Ayala A, Mulero V. 2007. Histamine is stored in mast cells of most evolutionarily advanced fish and regulates the fish inflammatory response. *Proc Natl Acad Sci U S A*
**104**:19434.

Reite OB. 1965. A phylogenetical approach to the functional significance of tissue mast cell histamine. *Nature*
**206**:1334.

Reite OB. 1972. Comparative physiology of histamine. Physiol Rev 52:778-819.

Takaya K, Fujita T, Endo K. 1967. Mast cells free of histamine in Rana catasbiana. *Nature*
**215**:776.

The following is the authors’ response to the original reviews.

**Reviewer #1 (Public Review):**
Summary:The global decline of amphibians is primarily attributed to deadly disease outbreaks caused by the chytrid fungus, Batrachochytrium dendrobatidis (Bd). It is unclear whether and how skin-resident immune cells defend against Bd. Although it is well known that mammalian mast cells are crucial immune sentinels in the skin and play a pivotal role in the immune recognition of pathogens and orchestrating subsequent immune responses, the roles of amphibian mast cells during Bd infections are largely unknown. The current study developed a novel way to enrich *X. laevis* skin mast cells by injecting the skin with recombinant stem cell factor (SCF), a KIT ligand required for mast cell differentiation and survival. The investigators found an enrichment of skin mast cells provides *X. laevis* substantial protection against Bd and mitigates the inflammation-related skin damage resulting from Bd infection. Additionally, the augmentation of mast cells leads to increased mucin content within cutaneous mucus glands and shields frogs from the alterations to their skin microbiomes caused by Bd.Strengths:This study underscores the significance of amphibian skin-resident immune cells in defenses against Bd and introduces a novel approach to examining interactions between amphibian hosts and fungal pathogens.

We thank the reviewer for acknowledging the novelty and importance of the work presented in our manuscript.

Weaknesses:The main weakness of the study is the lack of functional analysis of *X. laevis* mast cells. Upon activation, mast cells have the characteristic feature of degranulation to release histamine, serotonin, proteases, cytokines, and chemokines, etc. The study should determine whether X. laevis mast cells can be degranulated by two commonly used mast cell activators IgE and compound 48/80 for IgE-dependent and independent pathways. This can be easily done *in vitro*. It is also important to assess whether *in vivo* these mast cells are degranulated upon Bd infection using avidin staining to visualize vesicle releases from mast cells. Figure 3 only showed rSCF injection caused an increase in mast cells in naïve skin. They need to present whether Bd infection can induce mast cell increase and rSCF injection under Bd infection causes a mast cell increase in the skin. In addition, it is unclear how the enrichment of mast cells provides protection against Bd infection and alternations to skin microbiomes after infection. It is important to determine whether skin mast cells release any contents mentioned above.

We would like to thank the reviewer for taking the time to review our work and providing us with valuable feedback. We feel that we have successfully incorporated the reviewer’s suggestions into our revised manuscript, thereby improving this work.

Please note that amphibians do not possess the IgE antibody isotype (Flajnik, 2018).

To our knowledge there have been no published work assimilating approaches used when studying mammalian mast cell degranulation towards examining amphibian mast cells. While there are commercially available kits and reagents for examining mammalian mast cell granule content, most of these reagents do not cross-react with amphibian counterparts. This is especially true of cytokines and chemokines, which diverged quickly with evolution and thus do not share substantial protein sequence identity across species as diverged as frogs and mammals. Additionally, several studies suggest that amphibian mast cells lack histamine (Mulero et al., 2007; Reite, 1965, 1972; Takaya et al., 1967) and serotonin (Galli, 1990; Mulero et al., 2007). Respectfully, while following up on these findings is possible, we would not consider adopting approaches used in mammalian research to comparative immunology work as easy.

As noted in our manuscript, frog mast cells upregulate their expression of interleukin-4 (IL4), which is a hallmark cytokine associated with mammalian mast cells (Babina et al., 2016). The additional findings, presented in our revised manuscript indicate that mast cells respond to *Bd* by upregulating IL4 expression *in vitro* and *in vivo*. In turn, our work indicates that IL4 may be a central means by which frog mast cells confer protection against *Bd,* by counteracting *Bd*-elicited inflammation, including minimizing neutrophil infiltration, maintaining skin integrity, and promoting mucus production by skin mucus glands. Please find that these additional findings are presented in Figure 8 of our revised manuscript and are described in the results and discussion sections of the paper.

Our attempts to elicit degranulation of frog mast cells using compound 48/80 have so far not been successful. This may reflect technical issues with assays optimized for mammalian mast cells or biological difference between frog and mammalian mast cells, such as species differences in mas-related G-protein coupled receptors, through which compound 48/80 acts (Hermans et al., 2021). We will continue explore means to study frog mast cell degranulation both *in vitro* and *in vivo* but would also like to respectfully point out that while mast cell degranulation is a feature most associated with mammalian mast cells, this is not the only means by which the mammalian mast cells confer their immunological effects. Indeed, our additional studies suggest that mast cell IL4 production may be a key means by which these cells offer anti-*Bd* protection.

Please find that we have adopted an avidin-staining approach to visualize mast cell heparin content *in vitro* and to evaluate mast cell numbers *in vivo* in the skins of control and mast cell-enriched, mock- and *Bd*-infected animals. This additional work is depicted in Figure 4 of our revised manuscript and addressed in the results and discussion sections of our revised paper.

**Reviewer #2 (Public Review):**
Summary:In this study, Hauser et al investigate the role of amphibian (*Xenopus laevis*) mast cells in cutaneous immune responses to the ecologically important pathogen Batrachochytrium dendrobatidis (Bd) using novel methods of *in vitro* differentiation of bone marrow-derived mast cells and *in vivo* expansion of skin mast cell populations. They find that bone marrow-derived myeloid precursors cultured in the presence of recombinant *X. laevis* Stem Cell Factor (rSCF) differentiate into cells that display hallmark characteristics of mast cells. They inject their novel (r)SCF reagent into the skin of *X. laevis* and find that this stimulates the expansion of cutaneous mast cell populations *in vivo*. They then apply this model of cutaneous mast cell expansion in the setting of Bd infection and find that mast cell expansion attenuates the skin burden of Bd zoospores and pathologic features including epithelial thickness and improves protective mucus production and transcriptional markers of barrier function. Utilizing their prior expertise with expanding neutrophil populations in *X. laevis*, the authors compare mast cell expansion using (r)SCF to neutrophil expansion using recombinant colony-stimulating factor 3 (rCSF3) and find that neutrophil expansion in Bd infection leads to greater burden of zoospores and worse skin pathology.Strengths:The authors report a novel method of expanding amphibian mast cells utilizing their custom-made rSCF reagent. They rigorously characterize expanded mast cells *in vitro* and *in vivo* using histologic, morphologic, transcriptional, and functional assays. This establishes solid footing with which to then study the role of rSCF-stimulated mast cell expansion in the Bd infection model. This appears to be the first demonstration of the exogenous use of rSCF in amphibians to expand mast cell populations and may set a foundation for future mechanistic studies of mast cells in the *X. laevis* model organism.

We thank the reviewer for recognizing the breadth and extent of the undertaking that culminated in this manuscript. Indeed, this manuscript would not have been possible without considerable reagent development and adaptation of techniques that had previously not been used for amphibian immunity research. In line with the reviewer’s sentiment, to our knowledge this is the first report of using molecular approaches to augment amphibian mast cells, which we hope will pave the way for new areas of research within the fields of comparative immunology and amphibian disease biology.

Weaknesses:The conclusions regarding the role of mast cell expansion in controlling Bd infection would be stronger with a more rigorous evaluation of the model, as there are some key gaps and remaining questions regarding the data. For example:(1) Granulocyte expansion is carefully quantified in the initial time courses of rSCF and rCSF3 injections, but similar quantification is not provided in the disease models (Figures 3E, 4G, 5D-G). A key implication of the opposing effects of mast cell vs neutrophil expansion is that mast cells may suppress neutrophil recruitment or function. Alternatively, mast cells also express notable levels of csfr3 (Figure 2) and previous work from this group (Hauser et al, Facets 2020) showed rG-CSF-stimulated peritoneal granulocytes express mast cell markers including kit and tpsab1, raising the question of what effect rCSF3 might have on mast cell populations in the skin. Considering these points, it would be helpful if both mast cells and neutrophils were quantified histologically (based on Figure 1, they can be readily distinguished by SE or Giemsa stain) in the Bd infection models.

We thank the reviewer for this insightful suggestion. Please find that we successfully adopted an *in situ* hybridization approach to evaluate neutrophil numbers in the skins of control and mast cell-enriched, mock- and *Bd*-infected animals based on expression of the neutrophil marker, myeloperoxidase (*mpo*; (Buchan et al., 2019)). Please find these results are presented in Figures 6 and 8 of our revised manuscript and addressed in the appropriate sections of our revised paper.

Our findings suggest that rSCF administration results in the accumulation of mast cells that are polarized such, that they ablate the inflammatory response elicited by *Bd* infection, such as through mechanisms like IL4 production. Mammalian mast cells, including peritonea-resident mast cells, express *csf3r* (Aponte-Lopez et al., 2020; Jamur et al., 2010). For this reason, we used *mpo* expression to visualize neutrophil skin infiltration in Figures 6 and 8 of our revised work. While the *X. laevis* animal model does not permit nearly the degree of immune cell resolution afforded by mammalian animal models, we do know that the adult *X. laevis* peritonea contain a myriad of immune cell subsets. We anticipate that the high kit expression reported by

Hauser *et al.,* 2020 in the rCSF3-recruited peritoneal leukocytes reflects the presence of mast cells therein.Please find that we have used avidin-staining and *mpo in situ* hybridization to respectively visualize and enumerate mast cells and neutrophils in the skin of control and mast cell-enriched, mock- and *Bd*-infected animals. Indeed, our results show interesting, experimental condition-dependent changes in both the skin neutrophil and mast cell numbers. The results of these additional studies are presented in Figures 4, 6 and 8 of the revised manuscript and addressed in the results and discussions sections of our revised paper.

(2) Epithelial thickness and inflammation in Bd infection are reported to be reduced by rSCF treatment (Figure 3E, 5A-B) or increased by rCSF3 treatment (Figure 4G) but quantification of these critical readouts is not shown.

We thank the reviewer for this suggestion. We scored epithelial thickness under the distinct conditions described in our manuscript and presented the quantified data in Figures 5 and 8 of the revised paper.

(3) Critical time points in the Bd model are incompletely characterized. Mast cell expansion decreases zoospore burden at 21 dpi, while there is no difference at 7 dpi (Figure 3E). Conversely, neutrophil expansion increases zoospore burden at 7 dpi, but no corresponding 21 dpi data is shown for comparison (Figure 4G). Microbiota analysis is performed at a third time point,10 dpi (Figure 5D-G), making it difficult to compare with the data from the 7 dpi and 21 dpi time points. Reporting consistent readouts at these three time points is important to draw solid conclusions about the relationship of mast cell expansion to Bd infection and shifts in microbiota.

We thank reviewer for noting this discrepancy. Please find that we have repeated our mast cell-enrichment, *Bd*-challenge studies, examining days 10 and 21 post infection. Our new findings indicate that compared to control animals, mast cell-enrichment does result in significant reduction in *Bd* loads at both 10 and 21 dpi. The difference in Bd loads between r-ctrl and rSCF-treated animals at 10 dpi corroborates the other parameters that are altered between the two treatment groups at this experimental time point.

Our question regarding the roles of inflammatory granulocytes/neutrophils during Bd infections was that of ‘how’ rather ‘when’ these cells affect *Bd* infections. Thus, and because the central focus of this work was mast cells and not other granulocyte subsets; when we saw that rCSF3-recruited granulocytes adversely affect *Bd* infections at 7 days, we did not pursue the kinetics of these observations further. We plan to explore the roles of inflammatory mediators and immune cell subsets during the course of *Bd* infections but feel that these future studies are more peripheral to the central thesis of the present manuscript regarding the roles of frog mast cells during *Bd* infections.

(4) Although the effect of rSCF treatment on Bd zoospores is significant at 21 dpi (Figure 3E), bacterial microbiota changes at 21 dpi are not (Figure S3B-C). This discrepancy, how it relates to the bacterial microbiota changes at 10 dpi, and why 7, 10, and 21 dpi time points were chosen for these different readouts (Figure 5F-G), is not discussed.

Please find that our additional studies indicate that compared to control animals, frog skin mast cell-enrichment results in significant reduction in *Bd* loads at 10 dpi. This corroborate our other findings including the observation that at 10 dpi, control animals exhibit reduced microbial richness whereas mast cell-enriched frogs were protected from this disruption of their microbiome. The amphibian microbiome serves as a major barrier to these fungal infections (Walke and Belden, 2016) and we anticipate that *Bd*-mediated disruption of microbial richness facilitates host skin colonization by this pathogen. In turn, we anticipate that frog mast cells are conferring the observed anti-*Bd* protection in part by preventing microbial disassembly and thus interfering with optimal *Bd* colonization and growth on frog skins. Please find that we acknowledge and discuss these notions in our revised manuscript.

(5) The time course of rSCF or rCSF3 treatments relative to Bd infection in the experiments is not clear. Were the treatments given 12 hours prior to the final analysis point to maximize the effect? For example, in Figure 3E, were rSCF injections given at 6.5 dpi and 20.5 dpi? Or were treatments administered on day 0 of the infection model? If the latter, how do the authors explain the effects at 7 dpi or 21 dpi given mast cell and neutrophil numbers return to baseline within 24 hours after rSCF or rCSF3 treatment, respectively?

Please find that in our revised manuscript, we underlined the kinetics of our animal treatments and *Bd*-infections. In brief, for mast cell-enrichment, animals were injected with r-ctrl or rSCF, challenged 12 hours later with *Bd* and examined after 10 (per reviewers’ suggestions) and 21 days of infection. For neutrophil enrichment, animals were injected with r-ctrl or rCSF3, challenged 12 hours later with *Bd* and examined after 7 days of infection.

The title of the manuscript may be mildly overstated. Although Bd infection can indeed be deadly, mortality was not a readout in this study, and it is not clear from the data reported that expanding skin mast cells would ultimately prevent progression to death in Bd infections.

We acknowledge this point. The revised manuscript will be titled: “Amphibian mast cells: barriers to chytrid fungus infections”.

**Reviewer #3 (Public Review):**
Summary:Hauser et al. provide an exceptional study describing the role of resident mast cells in amphibian epidermis that produce anti-inflammatory cytokines that prevent Batrachochytrium dendrobatidis (Bd) infection from causing harmful inflammation, and also protect frogs from changes in skin microbiomes and loss of mucin in glands and loss of mucus integrity that otherwise cause changes to their skin microbiomes. Neutrophils, in contrast, were not protective against Bd infection. Beyond the beautiful cytology and transcriptional profiling, the authors utilized elegant cell enrichment experiments to enrich mast cells by recombinant stem cell factor, or to enrich neutrophils by recombinant colony-stimulating factor-3, and examined respective infection outcomes in *Xenopus*.Strengths:Through the use of recombinant IL4, the authors were able to test and eliminate the hypothesis that mast cell production of IL4 was the mechanism of host protection from Bd infection. Instead, impacts on the mucus glands and interaction with the skin microbiome are implicated as the protective mechanism. These results will press disease ecologists to examine the relative importance of this immune defense among species, the influence of mast cells on the skin microbiome and mucosal function, and open the potential for modulating mucosal defense.

We thank the reviewer for recognizing the utility of the work presented in our manuscript.

Weaknesses:A reduction of bacterial diversity upon infection, as described at the end of the results section, may not always be an "adverse effect," particularly given that anti-Bd function of the microbiome increased. Some authors (see Letourneau et al. 2022 ISME, or Woodhams et al. 2023 DCI) consider these short-term alterations as encoding ecological memory, such that continued exposure to a pathogen would encounter an enriched microbial defense. Regardless, mast cell-initiated protection of the mucus layer may negate the need for this microbial memory defense.

We thank the reviewer their insightful comment. We have revised our discussion to include this notion.

While the description of the mast cell location in the epidermal skin layer in amphibians is novel, it is not known how representative these results are across species ranging in chytridiomycosis susceptibility. No management applications are provided such as methods to increase this defense without the use of recombinant stem cell factor, and more discussion is needed on how the mast cell component (abundance, distribution in the skin) of the epidermis develops or is regulated.

We thank the reviewer for this suggestion. Please find that we have added a paragraph to our revised manuscripts to address possible source(s) of skin mast cells and a statement acknowledging that greater understanding of mast cell biology across distinct amphibian species may be used to develop future strategies for management of amphibian diseases.

We are very thankful to the reviewer for this excellent suggestion but would like to point out that the work presented in our manuscript was driven by comparative immunology questions more than by conservation biology. As such and considering just how little is known about mast cells outside of mammals; we chose not to speculate too much into possible utilities of altering amphibian skin mast cell composition and instead to focus our discussion on the immediate takeaways of the work presented by our paper.

**References**

Aponte-Lopez, A., Enciso, J., Munoz-Cruz, S., Fuentes-Panana, E.M., 2020. An *in Vitro* Model of Mast Cell Recruitment and Activation by Breast Cancer Cells Supports Anti-Tumoral Responses. *Int J Mol Sci*
**21**: 5293.

Babina, M., Guhl, S., Artuc, M., Zuberbier, T., 2016. IL-4 and human skin mast cells revisited: reinforcement of a pro-allergic phenotype upon prolonged exposure. *Arch Derm Re*s **308**: 665.

Buchan, K.D., Prajsnar, T.K., Ogryzko, N.V., de Jong, N.W.M., van Gent, M., Kolata, J., Foster, S.J., van Strijp, J.A.G., Renshaw, S.A., 2019. A transgenic zebrafish line for in vivo visualisation of neutrophil myeloperoxidase. *PLoS One*
**14**: e0215592.

Flajnik, M.F., 2018. A cold-blooded view of adaptive immunity. *Nat Rev Immunol* 18, 438-453.

Galli, S.J., 1990. New insights into "the riddle of the mast cells": microenvironmental regulation of mast cell development and phenotypic heterogeneity. *Lab Invest*
**62**: 5.

Hermans, M.A.W., van Stigt, A.C., van de Meerendonk, S., Schrijver, B., van Daele, P.L.A., van Hagen, P.M., van Splunter, M., Dik, W.A., 2021. Human Mast Cell Line HMC1 Expresses Functional Mas-Related G-Protein Coupled Receptor 2. *Front Immunol*
**12**, 625284.

Jamur, M.C., Moreno, A.N., Mello, L.F., Souza Junior, D.A., Campos, M.R., Pastor, M.V., Grodzki, A.C., Silva, D.C., Oliver, C., 2010. Mast cell repopulation of the peritoneal cavity: contribution of mast cell progenitors versus bone marrow derived committed mast cell precursors. *BMC Immunol*
**11**: 32.

Mulero, I., Sepulcre, M.P., Meseguer, J., Garcia-Ayala, A., Mulero, V., 2007. Histamine is stored in mast cells of most evolutionarily advanced fish and regulates the fish inflammatory response. *Proc Natl Acad Sci U S A*
**104**: 19434.

Reite, O.B., 1965. A phylogenetical approach to the functional significance of tissue mast cell histamine. *Nature*
**206**: 1334.

Reite, O.B., 1972. Comparative physiology of histamine. *Physiol Rev*
**52**: 778-819.

Takaya, K., Fujita, T., Endo, K., 1967. Mast cells free of histamine in Rana catasbiana. *Nature*
**215**: 776.

Walke, J.B., Belden, L.K., 2016. Harnessing the Microbiome to Prevent Fungal Infections: Lessons from Amphibians. *PLoS Pathog*
**12**: e1005796.

**Reviewer #2: (Recommendations For The Authors):**

We thank the reviewer for their excellent suggestions, their time reviewing this work and their help with this manuscript.

While we were not able to incorporate some of these changes, please find that we have significantly altered our manuscript in accordance with the reviewer’s suggestions from their public review. We feel that we have substantially altered our paper, including providing considerable additional data, supporting the key findings therein.

(1) The heatmap in Figure 1I appears to be scaled data, similar to Figure 4A, in which case the indicated scale numbers are not correct (e.g. they should be -2 to 2, or -3 to 3)

Thank you for the suggestion. Please find that we have changed this figure accordingly.

(2) For Figure 1, additional curated gene lists might better illustrate the difference in cell types, e.g. include the data for a panel of mast cell genes in a heatmap (mcpt1, tpsab1, etc.) and another panel of curated neutrophil genes (e.g. lyz) in a heatmap. If the authors still have leftover RNA, qPCR verification of some of the critical genes (e.g. kit) would add to the rigor of the analysis, as this study is the foundation of a new method for culturing amphibian mast cells.

We thank the reviewer for this suggestion. Unfortunately, we do not have leftover RNA/cDNA and we have not been able to locate mcpt1 or tpsab1 in our DEGs. We anticipate that this issue may stem from the suboptimal annotation of the *Xenopus laevis* genome. We agree that curating more mast cell/neutrophil genes would be ideal but feel that we have adequately highlighted those genes that are differentially expressed between the two populations in our analysis.

(3) The presentation of counts in Figure 2 is a bit hard to interpret. Although it is mentioned that everything is statistically significant, explicitly showing statistics for each gene would be better. One possibility would be to use a volcano plot (p-value vs log2 fold change) and highlight the genes shown in Figure 2, potentially with an accompanying heat map to show replicate variability.

We thank the reviewer for this suggestion. We entertained presenting the data as volcano plots or heat maps, but in the end felt that the bar graphs better conveyed the information that we are hoping to get across. Please note that the error bars in the bar graph depict the replicate variability. Please also note that to highlight that all the depicted genes were differentially expressed, we italicized the statement in the corresponding figure legend: “All depicted genes were significantly differentially expressed between the two populations”.

(4) Narratively, it might make more sense to put Figure 4A-C with Figure 3.

We thank the reviewer for this suggestion. Please find that we significantly revised most of our figures to better convey the content therein. We combined the content of Figure 4A-C with Figure 5A-C and added data on epidermal thickness under different conditions into this figure; Figure 5 of our revised manuscript.

(5) If possible, complementing the skin RNA-seq from rSCF treatment in Bd infection with skin RNA-seq from rCSF3 treatment to compare effects on transcriptional programs of barrier function, etc would elevate this study and add additional insights into cutaneous inflammation in the setting of Bd infection.

We thank the reviewer for this suggestion. We anticipate that the skin inflammation caused by *Bd* infection is not due solely to neutrophil infiltration and artificially altering the frog skin neutrophil content would thus not recapitulate chytridiomycosis progression. We completely agree that it would be valuable to examine barrier functions in control and mast cell-enriched, *Bd*-infected frogs. This is something that we hope to pursue further in future studies but feel that together with our additional findings, we are presenting a significant amount of data to constitute a stand-alone story.

(6) In Figure S1A, analyzing only 3 AMP genes by qPCR is perhaps too focused. As a control, it would be useful to also test some genes known to be functionally important in neutrophil anti-microbial responses, e.g. lyz. Expanding on this experiment by performing RNA-seq on Bd-treated, bone-marrow-derived mast cells and neutrophils would be a great addition to the manuscript and an important resource for future studies in the field. The fact that the use of rSCF (or rCSF3) enables the differentiation of these cells in large numbers of pure populations presents this unique opportunity. Although IL-4 did not end up affecting mucus production, clues to the mediator(s) of this mast cell-dependent effect may be found with unbiased RNA-seq after exposure to Bd.

We thank the reviewer for this suggestion but would like to point out that our manuscript is focused on mast cells rather than neutrophils. We also believe that in vitro exposure of leukocytes to *Bd* is not the most physiologically relevant model of what would happen to skin-resident and incoming immune cell subsets, since *Bd* primarily infects top-most keratinocytes. We anticipate that rather than coming into direct contact with the fungus, cells like mast cells and neutrophils are responding to Bd-produced and infected cell-produced products. For this reason, we did not perform RNA-seq analysis of in vitro derived mast cells or neutrophils stimulated with *Bd*. As we develop more *X. laevis*-specific reagents, we hope to revisit the question of infected skin mast cell and neutrophil gene expression profiles but are not in a position to ask these questions at this time.

This work is also guided by a finite budget, and we feel that together with our significant additional findings described in our revised manuscript, we are presenting a substantial amount of work to constitute a stand-alone story and manuscript.

**Reviewer #3 (Recommendations For The Authors):**
The following are minor edits needed in the text and figure legends:Standardize terms such as IL4 instead of il4 or ril4 vs rIL4 throughout. Also, r-SCF vs rSCF.

Thank you. Please find that we have standardized such terms throughout our revised manuscript. Please note that we are adhering to the convention that gene names are in lower case, protein names are in upper case and recombinant protein names are preceded by an ‘r’.

Pg 9 Change "In contract" to "In contrast".

Thank you and changed accordingly.

Fig 4 - Perhaps indicate if results in addition to 7dpi are also available.

Please find that we analyzed *Bd* loads in control and mast cell-enriched, infected frogs after 10 dpi. This data is presented in Figures 3 and 4 of our revised manuscript.

Similarly in Fig. 5, are results other than 10dpi available in the supplement?

Please find that the results from the microbiome studies are presented in supplemental figure 3 (Fig. S3). Please note that the results presented in original manuscript Fig. 5A-C - revised manuscript Fig. 5B-E depict data for 21 dpi, which is the longest examined infection timepoint. We present data from 1 and 10 dpi in Fig. 4 of our revised manuscript.

Indicate why these days were chosen in the methods.

Please find that we indicated why the experimental timepoints were chosen, in the methods section of our revised manuscript.

Fig S1 legend has errors in describing which panels are for which asterisks.Fig. S3 legend indicates panels F and G.

Thank you. Please find that we revised our supplemental figures and amended the corresponding figure legends.